# An Adaptive Half-Space Projection Method for Stochastic Optimization Problems with Group Sparse Regularization

**Yutong Dai**[*]                                                          *yud319@lehigh.edu*
*Department of Industrial and Systems Engineering*
*Lehigh University*

**Tianyi Chen**[*]                                                        *tiachen@microsoft.com*
*Microsoft*

**Guanyi Wang**                                                          *Guanyi.W@nus.edu.sg*
*Department of Industrial Systems Engineering and Management*
*National University of Singapore*

**Daniel P. Robinson**                                          *daniel.p.robinson@lehigh.edu*
*Department of Industrial and Systems Engineering*
*Lehigh University*

**Reviewed on OpenReview:** *https://openreview.net/forum?id=KBhSyBBeeO*

## Abstract

Optimization problems with group sparse regularization are ubiquitous in various popular downstream applications, such as feature selection and compression for Deep Neural Networks (DNNs). Nonetheless, the existing methods in the literature do not perform particularly well when such regularization is used in combination with a stochastic loss function. In particular, it is challenging to design a computationally efficient algorithm with a convergence guarantee and can compute group-sparse solutions. Recently, a half-space stochastic projected gradient (`HSPG`) method was proposed that partly addressed these challenges. This paper presents a substantially enhanced version of `HSPG` that we call `AdaHSPG+` that makes two noticeable advances. First, `AdaHSPG+` is shown to have a stronger convergence result under significantly looser assumptions than those required by `HSPG`. This improvement in convergence is achieved by integrating variance reduction techniques with a new adaptive strategy for iteratively predicting the support of a solution. Second, `AdaHSPG+` requires significantly less parameter tuning compared to `HSPG`, thus making it more practical and user-friendly. This advance is achieved by designing automatic and adaptive strategies for choosing the type of step employed at each iteration and for updating key hyperparameters. The numerical effectiveness of our proposed `AdaHSPG+` algorithm is demonstrated on both convex and non-convex benchmark problems. The source code is available at `https://github.com/tianyic/adahspg`.

## 1 Introduction

In many machine learning tasks, people not only want to compute solutions with small prediction/generalization errors but also seek easier-to-interpret models by identifying essential problem structures and filtering redundant parameters (Lin et al., 2019; Wen et al., 2016; Chen et al., 2020a; 2023b). One popular technique to achieve this goal is the use of structured sparsity-inducing regularization (Bach et al., 2012b; Jenatton et al., 2010) that encodes the model architecture. When this type of regularization is combined with an appropriate loss function, one is able to recover structured sparse solutions (Deleu & Bengio, 2021).

---

[*]These authors contributed equally to this work.

The non-overlapping group-sparse regularizer (Yuan & Lin, 2006) has proved useful in machine learning applications. It also plays an essential role for more general structured sparse learning tasks since problems with overlapping group sparsity (Bach et al., 2012a) or hierarchical sparsity (Jenatton et al., 2011) are often solved by converting them into an equivalent non-overlapping group-sparse problem (*e.g.*, by introducing latent variables (Bach et al., 2012b)). In general, structured sparsity has found numerous applications in computer vision (Elhamifar et al., 2012), signal processing (Chen & Selesnick, 2014), medical imaging (Liu et al., 2018), and deep learning (Scardapane et al., 2017; Li et al., 2020; Chen et al., 2021b; 2023a).

## 1.1 Problem Formulation

In this paper, we study the group-sparse regularization problem (sometimes called the mixed $\ell_1/\ell_2$ problem)

$$\min_{\boldsymbol{x} \in \mathbb{R}^n} \left\{ \psi(\boldsymbol{x}) := \underbrace{\frac{1}{N} \sum_{i=1}^{N} f_i(\boldsymbol{x})}_{f(\boldsymbol{x})} + \underbrace{\sum_{g \in \mathcal{G}} \lambda_g \left\| [\boldsymbol{x}]_g \right\|}_{r(\boldsymbol{x})} \right\} \tag{1}$$

where $\|\cdot\|$ denotes the $\ell_2$-norm, $f$ is the average of $N$ continuously differentiable functions $f_i : \mathbb{R}^n \to \mathbb{R}$, $\mathcal{G}$ is a set of disjoint subsets of $\mathcal{I} = \{1, 2, \ldots, n\}$ (each $g \in \mathcal{G}$ represents a group of variables), $\lambda_g > 0$ is the weight for group $g$, and $r(\boldsymbol{x})$ is the mixed $\ell_1/\ell_2$ norm [*]. Larger values for $\lambda_g$ result in greater group sparsity but may produce more model bias. In practice, $\lambda_g$ is usually tuned to achieve an acceptable balance between sparsity and model generalizability.

## 1.2 Literature review

Problem (1) has been well studied in deterministic optimization, and there are many algorithms capable of returning sparse approximate solutions (Yuan & Lin, 2006; Roth & Fischer, 2008; Huang et al., 2011; Ndiaye et al., 2017). Proximal methods are popular approaches for solving the structured non-smooth optimization problem (1). These methods include the proximal-gradient (PG) method, which only uses first-order derivatives. When $N$ is huge, stochastic methods become essential because they use small subsets of the data at each iteration, whereas deterministic methods perform costly evaluations using the entire dataset.

The proximal stochastic gradient method (`Prox-SG`) (Duchi & Singer, 2009) is the natural (stochastic) extension of the PG method. The regularized dual-averaging method (`RDA`) (Xiao, 2010; Yang et al., 2010) applies the dual averaging scheme from (Nesterov, 2009) to `Prox-SG`. There exists a set of incremental gradient methods inspired by stochastic average gradient (`SAG`) (Roux et al., 2012) that utilize the average of accumulated past gradients to improve the convergence rate. For example, the proximal stochastic variance-reduced gradient method (`Prox-SVRG`) (Xiao & Zhang, 2014) and proximal spider (`Prox-Spider`) (Zhang & Xiao, 2019) use multi-stage schemes based on the well-known variance reduction technique proposed in SVRG (Johnson & Zhang, 2013) and Spider (Fang et al., 2018). `SAGA` (Defazio et al., 2014) is a blend of ideas from `SAG` and `Prox-SVRG`.

Compared to deterministic methods, the above state-of-the-art stochastic algorithms for solving problem (1) have weaknesses (Chen et al., 2021a;b). In particular, these stochastic algorithms typically struggle to achieve near-optimal solutions that are also group sparse because of the randomness in the algorithm and the limitations in ensuring sparsity.

To address the drawbacks of stochastic proximal methods, a two-stage algorithm `HSPG` (half-space stochastic projected gradient method) was recently proposed with the aim of improving numerical performance, specifically to improve solution sparsity (Chen et al., 2020b; 2021b). In the first stage, a subgradient algorithm is run to seek an estimate of a solution. The second stage uses the solution estimate from the first stage to define a certain half-space that is used to improve the sparsity of the solution returned in the first stage. Although `HSPG` was shown to perform well in initial testing, it has limitations. First, since `HSPG` is a two-stage method, it is unable to converge when an insufficiently accurate solution estimate is provided

---

[*]The name $\ell_1/\ell_2$ norm stems from the fact it can set the norm of some subvectors (indexed by induced $g$) to zero, hence is equivalent to selecting a subset of a group of variables that are nonzero.

from the first stage. This is not ideal from both a theoretical and numerical performance perspective since much computational effort is likely wasted in stage one in an attempt to ensure an accurate enough solution estimate. Second, the analysis provided for HSPG does not extend to the mixed $\ell_1/\ell_2$ problem since it is not Lipschitz continuous in a neighborhood of the origin. Third, the numerical performance of HSPG relies heavily on tuning a couple of crucial hyperparameters, which is typically quite time-consuming. The focus of our research is to address the weaknesses of the HSPG approach, as we describe in the next section.

## 1.3 Our contributions

We overcome the weaknesses of HSPG discussed above by proposing an *adaptive* half-space stochastic projected gradient method (AdaHSPG+). Our main contributions can be summarized as follows.

- **Improved algorithmic design and convergence theory.** Our proposed approach AdaHSPG+ is a significance improvement upon HSPG. First, unlike HSPG, our AdaHSPG+ is not a two-stage method and therefore does not require a "good enough" solution estimate to be provided from a first stage. Second, although we use a similar half-space projection scheme to enhance solution sparsity, our strategy for choosing the groups that define the half-space is significantly different from that used in HSPG. Consequently, unlike HSPG, we establish a convergence theory that holds for problems that are not Lipschitz continuous in a neighborhood of zero, such as the group sparse problem (1) considered in this paper. Even though the HSPG theory does not hold in our setting, it is interesting to note that the theory provided for HSPG (for other types of regularizers) is an expectation-type result that holds with a certain probability. In contrast, the convergence theory we present is a pure expectation-type result akin to standard results for stochastic gradient methods. The key to achieving these improvements over HSPG is an automatic and adaptive strategy that decides the iterations for which the half-space projection scheme should be used.

- **Numerical performance.** We numerically demonstrate the effectiveness of AdaHSPG+ on commonly tested convex (logistic regression) and nonconvex (deep neural networks (DNNs)) problems. In the convex setting, AdaHSPG+ performs similarly to HSPG and significantly outperforms Prox-SG and Prox-SVRG. For the nonconvex setting, AdaHSPG+ significantly outperforms all of the methods, including HSPG, especially in terms of solution sparsity. These numerical gains are due to the combination of the enhancements described above and a new automatic and adaptive strategy for defining a key hyperparameter that defines the half-space projections.

## 1.4 Preliminaries and notations

Unless specified otherwise, $\|\cdot\|$ represents the $\ell_2$ norm, and $|\mathcal{A}|$ denotes the cardinality of a set $\mathcal{A}$. $[N]$ is the index set $\{1, 2, \cdots, N\}$ and $\mathbb{N}_+$ and $\mathbb{R}_+$ are the set of positive integers and positive real numbers, respectively. For a nonempty set $\mathcal{B} \subseteq [N]$, we define the batch function induced by the set $\mathcal{B}$

$$f_{\mathcal{B}}(\boldsymbol{x}) := \frac{1}{|\mathcal{B}|} \sum_{i \in \mathcal{B}} f_i(\boldsymbol{x}) \text{ and } \psi_{\mathcal{B}}(\boldsymbol{x}) := f_{\mathcal{B}}(\boldsymbol{x}) + r(\boldsymbol{x}).$$

For any vector $\boldsymbol{x} \in \mathbb{R}^n$ and $g \in \mathcal{G}$, $[\boldsymbol{x}]_g \in \mathbb{R}^{|g|}$ is the subvector obtained by restricting $\boldsymbol{x}$ to the elements in $g$. For any $[\boldsymbol{x}]_g \neq 0$, the partial gradient of $\psi$ over $[\boldsymbol{x}]_g$ is defined as $\nabla_g \psi_{\mathcal{B}}(\boldsymbol{x}) = \nabla_g f_{\mathcal{B}}(\boldsymbol{x}) + \nabla_g r(\boldsymbol{x})$. In other words, $\nabla_g \psi_{\mathcal{B}}(\boldsymbol{x}) = \partial \psi_{\mathcal{B}}(\boldsymbol{x})/\partial [\boldsymbol{x}]_g$. For any $\eta > 0$, the proximal operator induced by $r$ is defined as $\mathbf{Prox}_{\eta r}(\boldsymbol{x}) := \arg\min_{\boldsymbol{u} \in \mathbb{R}^n} \frac{1}{2\eta} \|\boldsymbol{u} - \boldsymbol{x}\|_2^2 + r(\boldsymbol{u})$, which has a closed-form solution (Beck, 2017, Example 6.19), *i.e.*, $[\mathbf{Prox}_{\eta r}(\boldsymbol{x})]_g = \left(1 - \frac{\alpha}{\max\{\|\boldsymbol{x}_g\|, \alpha\}}\right)[\boldsymbol{x}]_g$. Furthermore, we define the proximal gradient step over the index set $\mathcal{B}$ as

$$\mathbf{s}_{\mathcal{B}}(\boldsymbol{x}; \eta) = \mathbf{Prox}_{\eta r}(\boldsymbol{x} - \eta \nabla f_{\mathcal{B}}(\boldsymbol{x})) - \boldsymbol{x}, \tag{2}$$

and when $\mathcal{B} = [N]$, we simply write $\mathbf{s}(\boldsymbol{x}; \eta)$ and $\mathbf{s}(\boldsymbol{x}; \eta)$ is the negative of gradient mapping defined in (Beck, 2017, Definition 10.5) The quantity $\|\mathbf{s}(\boldsymbol{x}; \eta)\|$ is a stationarity measure for problem (1) (Beck, 2017, Theorem 10.7). Since solution sparsity is of great interest in this paper, we define the concept of support.

**Definition 1.1.** The support of a point $\boldsymbol{x} \in \mathbb{R}^n$ with respect to $\mathcal{G}$ is $\mathrm{Supp}(\boldsymbol{x}) := \{g \in \mathcal{G} \mid [\boldsymbol{x}]_g \neq 0\}$.

## 2 `AdaHSPG+` **Algorithm**

In this section, we propose an Adaptive Half-Space Stochastic Projected Gradient (`AdaHSPG+`) method for solving problem (1) that uses optimal support prediction, space decomposition, and half-space projections. An overview of the algorithm is presented in Section 2.1. For every epoch, either an `Enhanced Half-Space step` or a `Prox-SVRG step` is performed based on an automatic switching mechanism. In general, the `Prox-SVRG step` guides how we estimate the solution support (the non-zero groups of variables), and the `Enhanced Half-Space step` is designed to improve the sparsity of the iterates. These two steps work together along with the switching mechanism to drive the convergence of the iterates generated by `AdaHSPG+`.

### 2.1 Main algorithm

---
**Algorithm 1** `AdaHSPG+` for solving (1).

---
1: **Input:** $\boldsymbol{x}_0 \in \mathbb{R}^n$, $\mu > 0$, $\epsilon \in (0, 1)$, $(\alpha_0, \eta, \kappa) \in \mathbb{R}_+^3$, $(m_h, m_p, b) \in \mathbb{N}_+^3$.
2: **for** epoch $k = 0, 1, 2, \ldots$ **do**
3:      Randomly sample an index set $\mathcal{B}_k \subseteq [N]$, and then compute $\nabla f_{\mathcal{B}_k}(\boldsymbol{x}_k)$ and step $s_{\mathcal{B}_k}(\boldsymbol{x}_k, \eta)$ as in (2).
4:      Compute $\mathcal{I}_k^{\mathtt{HS}}$ and $\mathcal{I}_k^{\mathtt{NHS}}$ using (4).
5:      Compute the measures $\chi_k^{\mathtt{HS}} = \|[s_{\mathcal{B}_k}(\boldsymbol{x}_k, \eta)]_{\mathcal{I}_k^{\mathtt{HS}}}\|$ and $\chi_k^{\mathtt{NHS}} = \|[s_{\mathcal{B}_k}(\boldsymbol{x}_k, \eta)]_{\mathcal{I}_k^{\mathtt{NHS}}}\|$ .
6:      **if** $\chi_k^{\mathtt{HS}} \geq \mu \chi_k^{\mathtt{NHS}}$ **then**
7:          Call Algorithm 2 to get the next iterate: $\boldsymbol{x}_{k+1} \leftarrow \text{EHS}(\boldsymbol{x}_k, \alpha_k, \epsilon, \kappa, m_h)$.
8:      **else**
9:          Call Algorithm 3 to get the next iterate: $\boldsymbol{x}_{k+1} \leftarrow \text{PSVRG}(\boldsymbol{x}_k, \nabla f_{\mathcal{B}_k}(\boldsymbol{x}_k), \eta, m_p, b)$.
10:      **end if**
11:      Set $\alpha_{k+1}$ (*e.g.*, constant or diminishing step size).
12: **end for**

---

At the beginning of the $k$th epoch, we partition the groups of variables into two sets. To form these sets, let us choose a constant $\kappa \in (0, \infty)$ and define the function

$$\mathcal{I}(\boldsymbol{x}, \mathcal{B}) := \left\{ g \in \mathcal{G} \ \middle| \ \|[\boldsymbol{x}]_g\| \neq 0, [\boldsymbol{x} + \mathbf{s}_{\mathcal{B}}(\boldsymbol{x}, \eta)]_g \neq 0, \frac{\|[\boldsymbol{x}]_g\|}{\|\nabla_g \psi_{\mathcal{B}}(\boldsymbol{x})\|} \geq \kappa \right\} \tag{3}$$

which we use to define the sets

$$\mathcal{I}_k^{\mathtt{HS}} = \mathcal{I}(x_k, \mathcal{B}_k) \quad \text{and} \quad \mathcal{I}_k^{\mathtt{NHS}} = \mathcal{G} \setminus \mathcal{I}_k^{\mathtt{HS}}. \tag{4}$$

These sets aid in identifying the zero/nonzero groups at a solution. In particular, the groups of variables in $\mathcal{I}_k^{\mathtt{HS}}$ are input to the `Enhanced Half-Space step` computation to explore further whether they should remain nonzero. This makes sense since $g \in \mathcal{I}_k^{\mathtt{HS}}$ for the given $x_k$ and minibatch $\mathcal{B}_k$ if and only if the following conditions hold: (i) $x_k$ is nonzero over the group $g$ (*i.e.*, $\|[\boldsymbol{x}_k]_g\| \neq 0$); (ii) the batch proximal gradient step over $\mathcal{B}_k$ predicts that group $g$ is in the solution support (i.e, $[\boldsymbol{x}_k + s_{\mathcal{B}_k}(\boldsymbol{x}_k, \eta)]_g \neq 0$), and (iii) the restriction of $x_k$ to group $g$ is sufficiently far from the origin (*i.e.*, $\|[\boldsymbol{x}_k]_g\| \geq \kappa \|\nabla_g \psi_{\mathcal{B}_k}(\boldsymbol{x}_k)\|$), which is needed for our convergence analysis.

The *switching mechanism* that we have designed uses $\chi_k^{\mathtt{HS}}$ and $\chi_k^{\mathtt{NHS}}$ in line 5 of Algorithm 1 to decide which one the two spaces of variables we should currently explore further. This makes sense since the sizes of $\chi_k^{\mathtt{HS}}$ and $\chi_k^{\mathtt{NHS}}$ indicate the amount of progress one might expect to achieve by continuing to optimize over the variables in $\mathcal{I}_k^{\mathtt{HS}}$ (`Enhanced Half-Space step`) and $\mathcal{I}_k^{\mathtt{NHS}}$ (`Prox-SVRG step`), respectively. If $\chi_k^{\mathtt{HS}} \geq \mu \chi_k^{\mathtt{NHS}}$, the algorithm proceeds with the `Enhanced Half-Space step`; otherwise, the algorithm performs the `Prox-SVRG step` and then generates a new partition of the groups. The value chosen for $\mu$ gives preference to either the `Enhanced Half-Space step` or `Prox-SVRG step`.

## 2.2 An enhanced half-space step

---
**Algorithm 2** Enhanced Half-Space step (EHS).

---
1: **Input:** $\boldsymbol{x}_k \in \mathbb{R}^n$, $\alpha_k > 0$, $\epsilon \in (0,1)$, $m_h > 0$, and $\kappa > 0$.
2: Initialize $\tilde{\boldsymbol{x}}_{k,0} = \boldsymbol{x}_k$.
3: **for** $t = 0, \cdots, m_h - 1$ **do**
4:  Randomly sample an index set $\mathcal{B}_{k,t} \subseteq [N]$.
5:  Form $\mathcal{I}_{k,t}^{\text{HS}} = \mathcal{I}(\tilde{\boldsymbol{x}}_{k,t}, \mathcal{B}_{k,t})$ and $\mathcal{I}_{k,t}^{\text{NHS}} = \mathcal{G} \setminus \mathcal{I}_{k,t}^{\text{HS}}$.
6:  Compute $\nabla_{\mathcal{I}_{k,t}^{\text{HS}}} \psi_{\mathcal{B}_{k,t}}(\tilde{\boldsymbol{x}}_{k,t})$, which is well defined.
7:  Compute a trial iterate $\hat{\boldsymbol{x}}_{k,t+1}$ as
8:  $\qquad [\hat{\boldsymbol{x}}_{k,t+1}]_{\mathcal{I}_{k,t}^{\text{HS}}} \leftarrow [\tilde{\boldsymbol{x}}_{k,t}]_{\mathcal{I}_{k,t}^{\text{HS}}} - \alpha_k \nabla_{\mathcal{I}_{k,t}^{\text{HS}}} \psi_{\mathcal{B}_{k,t}}(\tilde{\boldsymbol{x}}_{k,t}), \qquad [\hat{\boldsymbol{x}}_{k,t+1}]_{\mathcal{I}_{k,t}^{\text{NHS}}} \leftarrow [\tilde{\boldsymbol{x}}_{k,t}]_{\mathcal{I}_{k,t}^{\text{NHS}}}.$
9:  Set $\tilde{\boldsymbol{x}}_{k,t+1} \leftarrow \textbf{Proj}_{\mathcal{I}_{k,t}^{\text{HS}}}(\hat{\boldsymbol{x}}_{k,t+1}; \tilde{\boldsymbol{x}}_{k,t})$ using (5).
10: **end for**
11: **return** $\boldsymbol{x}_{k+1} \leftarrow \tilde{\boldsymbol{x}}_{k,m_h}$

---

If the `Enhanced Half-Space step` (EHS) is computed during the $k$th epoch of `AdaHSPG+`, then the goal is to explore group sparsity with respect to the current candidate support set. Specifically, in Algorithm 2, we form the partial stochastic gradient of $\psi$ over groups of variables in $\mathcal{I}_{k,t}^{\text{HS}}$, where $\mathcal{I}_{k,t}^{\text{HS}} = \mathcal{I}(\tilde{\boldsymbol{x}}_{k,t}, \mathcal{B}_{k,t})$, $\tilde{\boldsymbol{x}}_{k,t}$ and $\mathcal{B}_{k,t}$ are the $t$th iterate and mini-batch of Algorithm 2, and $\mathcal{I}(\cdot, \cdot)$ is defined in (3), and then take a stochastic gradient step (see line 8 in Algorithm 2) to obtain $\hat{\boldsymbol{x}}_{k,t+1}$. For $t = 0$, the gradient of $f$ computed over $\mathcal{B}_k$ as part of the switching mechanism in Algorithm 1 can be reused. Next, a half-space projection operator

$$[\textbf{Proj}_{\mathcal{I}}(\boldsymbol{z}; \boldsymbol{x})]_g = \begin{cases} \boldsymbol{0} & \text{if } [\boldsymbol{x}]_g^T [\boldsymbol{z}_g] \le \epsilon \left\| [\boldsymbol{x}]_g \right\|^2 \text{ and } g \in \mathcal{I} \\ [\boldsymbol{z}]_g & \text{otherwise} \end{cases} \tag{5}$$

is performed on $\hat{\boldsymbol{x}}_{k,t+1}$ to obtain $\tilde{\boldsymbol{x}}_{k,t+1}$. A graphical illustration of the projection procedure is presented in Figure 1. The intuition behind the above projection is to project groups of variables to zero when the stochastic gradient step crosses over the boundary of the hyperplane $\mathcal{H}_g = \{\boldsymbol{z}_g \in \mathbb{R}^{|g|} \mid [\boldsymbol{x}]_g^T \boldsymbol{z}_g - \epsilon \left\| [\boldsymbol{x}]_g \right\|^2 = 0\}$ depicted in Figure 1. The parameter $\epsilon$ in (5) determines the distance between the boundary of $\mathcal{H}_g$ and the origin, and therefore controls how aggressively sparsity is sought. Larger values of $\epsilon$ are more likely to lead to solutions with greater group sparsity. For convenience, we let $\mathcal{I}_{k,t}^{\text{PROJ}}$ as the index set of the groups of variables that are projected onto zero via (5) and $\mathcal{I}_{k,t}^{\text{GRAD}} = \mathcal{I}_{k,t}^{\text{HS}} \setminus \mathcal{I}_{k,t}^{\text{PROJ}}$ as the index set for the remaining complementary groups in $\mathcal{I}_{k,t}^{\text{HS}}$.

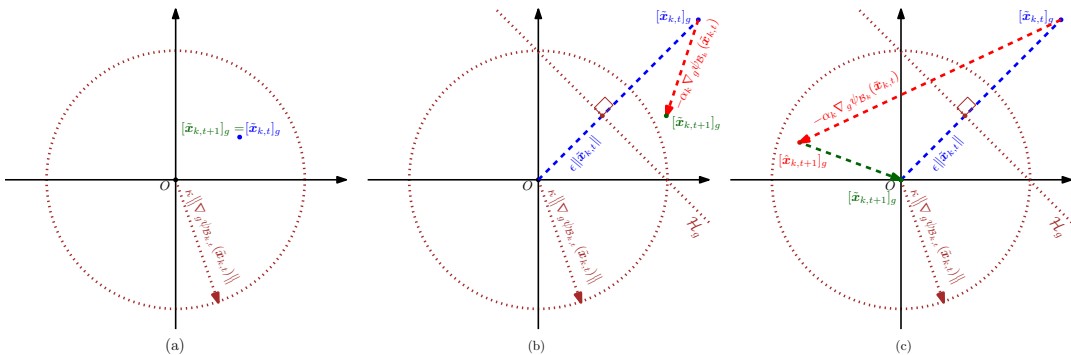

Figure 1: An illustration of the proposed half-space projection. ($a$) The restriction of the current iterate to group $g$, namely $[\tilde{\boldsymbol{x}}_{k,t}]_g$, is not sufficiently far from the origin. The `Enhanced Half-Space step` does not perform an update. ($b$) Here, $[\tilde{\boldsymbol{x}}_{k,t}]_g$ is sufficiently far from the origin, and a stochastic gradient step does not cross the boundary of the hyperplane $\mathcal{H}_g$. The projection leaves $[\tilde{\boldsymbol{x}}_{k,t+1}]_g$ unchanged. ($c$) Here, $[\tilde{\boldsymbol{x}}_{k,t}]_g$ is sufficiently far from the origin, and the stochastic gradient step crosses the boundary of $\mathcal{H}_g$. The projection sets $[\tilde{\boldsymbol{x}}_{k,t+1}]_g = 0$, thus creating a sparser solution estimate.

### 2.3 A Prox-SVRG step

When Algorithm 1 chooses to compute the `Prox-SVRG step` during the $k$th epoch, we simply perform $m_p$ updates of a basic `Prox-SVRG` method as described in Algorithm 3.

---

**Algorithm 3** `Prox-SVRG step` (PSVRG).

---

1: **Input:** $\boldsymbol{x}_k \in \mathbb{R}^n$, $\eta > 0$, $m_p > 0$, and $\nabla f_{\mathcal{B}_k}(\boldsymbol{x}_k)$.
2: Set $\tilde{\boldsymbol{x}}_{k,0} \leftarrow \boldsymbol{x}_k$.
3: **for** $t = 0, \cdots, m_p - 1$ **do**
4:      Randomly sample an index set $\mathcal{B}_{k,t} \subseteq [N]$ such that $|\mathcal{B}_{k,t}| = b$.
5:      $\boldsymbol{v}_{k,t} \leftarrow \frac{1}{b} \sum_{i \in \mathcal{B}_{k,t}} (\nabla f_i(\tilde{\boldsymbol{x}}_{k,t}) - \nabla f_i(\boldsymbol{x}_k)) + \nabla f_{\mathcal{B}_k}(\boldsymbol{x}_k)$
6:      $\tilde{\boldsymbol{x}}_{k,t+1} \leftarrow \mathbf{Prox}_{\eta r}(\tilde{\boldsymbol{x}}_{k,t} - \eta \boldsymbol{v}_{k,t})$
7: **end for**
8: **return** $\boldsymbol{x}_{k+1} \leftarrow \tilde{\boldsymbol{x}}_{k,m_p}$

---

`Prox-SVRG` extends `Prox-SG` by adding a variance reduction technique (see Johnson & Zhang (2013); Xiao & Zhang (2014)), which has been frequently studied in J Reddi et al. (2016); Li & Li (2018). Compared to `Prox-SG`, `Prox-SVRG` enjoys a stronger convergence theory and often performs better on convex experiments, although this numerical advantage often disappears on non-convex problems such as those in various DNN applications (Defazio & Bottou, 2019).

Although the proximal gradient operator is known to identify the solution support (asymptotically and under certain assumptions) in the deterministic setting, the story is different in the stochastic regime. In fact, stochastic methods applied in our setting typically return dense solution estimates (*e.g.*, see Chen et al. (2021b)). In `AdaHSPG+`, however, the `Prox-SVRG step` is complemented by the `Enhanced Half-Space step` to promote sparse solutions.

## 3 Convergence Analysis

In this section, we describe the convergence results for the proposed `AdaHSPG+` algorithm; all proof may be found in the appendix. We assume the following hold throughout.

**Assumption 3.1.** The function $f_i : \mathbb{R}^n \to \mathbb{R}$ is continuously differentiable and has an $L$-Lipschitz continuous gradient for all $i \in \{1, 2, \cdots, N\}$, *i.e.*, $f_i$ is $L$-smooth. It follows that $f$ is also $L$-smooth.

**Assumption 3.2.** The function $\psi$ is bounded below, *i.e.*, $\psi(\boldsymbol{x}) > \underline{\psi}$ for all $\boldsymbol{x} \in \mathbb{R}^n$ and some $\underline{\psi} \in \mathbb{R}$.

**Assumption 3.3.** The step size sequence $\{\alpha_k\}$ satisfies $\sum_{k=0}^{\infty} \alpha_k = \infty$ and $\sum_{k=0}^{\infty} \alpha_k^2 < \infty$.

**Assumption 3.4.** The iterate sequence $\{\boldsymbol{x}_k\}$ is bounded, *i.e.*, $\|\boldsymbol{x}_k\| \leq M$ for all $k$ and some $M > 0$.

Assumption 3.1 and Assumption 3.2 are standard in the literature (Xiao & Zhang, 2014; Johnson & Zhang, 2013; Rosasco et al., 2019). Assumption 3.3 and Assumption 3.4 are used in (Ljung, 1977) to analyze the convergence behavior of stochastic gradient descent. Assumption 3.4 is reasonable in our setting since the regularizer penalizes the magnitude of variables, and $\{\|\boldsymbol{x}_k\|\}$ remained bounded in our numerical tests.

We can prove that the `Enhanced Half-Space step` has the following sufficient decrease property.

**Lemma 3.5.** *If the* `Enhanced Half-Space step` *is computed during the $k$th epoch of Algorithm 1, then the $t$th iterate $\tilde{\boldsymbol{x}}_{k,t+1}$ computed in Algorithm 2 satisfies*

$$
\mathbb{E}_{\mathcal{B}_{k,t}}[\psi(\tilde{\boldsymbol{x}}_{k,t+1}) \mid \tilde{\boldsymbol{x}}_{k,t}] \leq \psi(\tilde{\boldsymbol{x}}_{k,t}) - \left(\frac{1-\epsilon}{\alpha_k} - \frac{L}{2}\right) \left\|[\tilde{\boldsymbol{x}}_{k,t}]_{\mathcal{I}_{k,t}^{PROJ}}\right\|^2 - \alpha_k \left(1 - \frac{L\alpha_k}{2}\right) \left\|\nabla_{\mathcal{I}_{k,t}^{GRAD}} \psi(\tilde{\boldsymbol{x}}_{k,t})\right\|^2
$$
$$
+ \frac{\lambda_{\max} \alpha_k^2}{\kappa^2 \epsilon} \sum_{g \in \mathcal{I}_{k,t}^{GRAD}} \|[\tilde{\boldsymbol{x}}_{k,t}]_g\| + \frac{L\alpha_k^2 \sigma^2 \mathbb{I}(|\mathcal{B}_{k,t}| < N)}{2|\mathcal{B}_{k,t}|},
$$

*where the expectation is taken with respect to $\mathcal{B}_{k,t}$ and conditioning on $\tilde{\boldsymbol{x}}_{k,t}$. $\mathbb{I}(\cdot)$ is the indicator function, and $\lambda_{\max} = \max\{\lambda_g \mid g \in \mathcal{G}\}$. Furthermore, $\sigma^2$ is the bounded variance constant satisfying $\mathbb{E}\|\nabla f_i(x) - \nabla f(x)\| \leq$*

$\sigma^2$, *with the expectation taken with respect to randomly sampled $i$. (The existence of the constant $\sigma^2$ follows from the finite-sum structure of $f(x)$, the uniform sampling scheme, and Assumptions 3.1 and 3.4.)*

Lemma 3.5 shows that for step size $\alpha_k$ sufficiently small and $|\mathcal{B}_{k,t}|$ sufficiently large, the `Enhanced Half-Space step` is expected to decrease the objective function $\psi$.

For completeness, we present the sufficient decrease property of `Prox-SVRG` from (Li & Li, 2018, pg. 14).

**Lemma 3.6.** *If the `Prox-SVRG step` is computed during the $k$th epoch of Algorithm 1 and $|\mathcal{B}_k| \leq N$, $|\mathcal{B}_{k,t}| = \min(m_p^2, N)$, $m_p(m_p - 1/2) \leq b$, and $\eta \leq \frac{1}{6L}$, then*

$$\mathbb{E}[\psi(\boldsymbol{x}_k)] \leq \ \mathbb{E}\left[\psi(\boldsymbol{x}_{k-1}) - \sum_{t=0}^{m_p-1} \frac{1}{36L\eta^2} \|\mathbf{s}(\tilde{\boldsymbol{x}}_{k,t};\eta)\|^2 + \sum_{t=0}^{m_p-1} \frac{\mathbb{I}(|\mathcal{B}_k| < N)\eta\sigma^2}{|\mathcal{B}_k|}\right].$$

Lemma 3.6 shows that when the batch size $\mathcal{B}_k$ is sufficiently large, the `Prox-SVRG step` is expected to yield a decrease in the objective function $\psi$.

We are now ready to state the main convergence theorem.

**Theorem 3.7.** *Under Assumptions 3.1-3.4, and with $\epsilon \in (0,1)$, $\eta \in (0, \frac{1}{6L}]$, $\alpha_k \leq \min\left\{\frac{2(1-\epsilon)}{L}, 1\right\}$, $|\mathcal{B}_k| = N$ for all $k$, $m_p(m_p - 1/2) \leq b$, and $|\mathcal{B}_{k,t}| = \min(m_p^2, N)$ for all $k$ and $t$, then Algorithm 1 generates a sequence of iterates $\{\boldsymbol{x}_k\}_{k\in\mathbb{N}}$ such that $\liminf_{k\to\infty} \mathbb{E}[\|\mathbf{s}(\boldsymbol{x}_k;\eta)\|] = 0$.*

Theorem 3.7 shows that if the sequence $\{\boldsymbol{x}_k\}_{k\in\mathbb{N}}$ computed by Algorithm 1 has a limit point, then that limit point will be a stationary point in expectation. We remark that this theorem only establishes the asymptotic convergence result, whereas the previous works (Li & Li, 2018; Pham et al., 2020) establish the complexity bounds. The main challenge that prevents us from establishing such a convergence result is the diminishing stepsize used in the `Enhanced Half-Space step`. We conjecture that with additional variance reduction integrated into the `Enhanced Half-Space step` one might be able to establish such a result. Here, we avoid the variance reduction technique in the `Enhanced Half-Space step` because the practical performance on deep learning tasks is significantly better.

We then reveal the sparsity identification property of `AdaHSPG+` as stated in Theorem 3.8, which requires a non-degeneracy assumption, which is similar to (Chen et al., 2018; Nutini et al., 2019). Define

$$0 < \delta := \frac{1}{2} \min_{g\in\mathcal{I}^0(\boldsymbol{x}^*)} \inf_{\mathcal{B}\subseteq[n]} \left(\lambda - \|[\nabla f_{\mathcal{B}}(\boldsymbol{x}^*)]_g\|\right). \tag{6}$$

**Theorem 3.8.** *Given any $k \in \mathbb{N}_+$, if the $k$th epoch performs an `Enhanced Half-Space step` with $\alpha_k \leq 1/L$, then for any $t \in \{0, \ldots, m_h - 1\}$ satisfying $\|\tilde{\boldsymbol{x}}_{x,t} - \boldsymbol{x}^*\| \leq \frac{2\alpha_k\delta}{1-\epsilon+\alpha_k L}$, $[\tilde{\boldsymbol{x}}_{k,t} + \mathbf{s}(\tilde{\boldsymbol{x}}_{k,t};\eta)]_{\mathcal{I}^0(\boldsymbol{x}^*)\cap\mathcal{I}^{\neq 0}(\tilde{\boldsymbol{x}}_{k,t})} \neq 0$, and $\kappa \leq \min_{g\in\mathcal{I}^0(\boldsymbol{x}^*)} \|[\tilde{\boldsymbol{x}}_{k,t}]_g\| / \|[\nabla\psi_{\mathcal{B}_{k,t}}(\tilde{\boldsymbol{x}}_{k,t})]_g\|$, it holds that `AdaHSPG+` yields $\mathcal{I}^0(\boldsymbol{x}^*) \subseteq \mathcal{I}^0(\tilde{\boldsymbol{x}}_{k,t+1})$, where $\mathcal{I}^0(\boldsymbol{x})$ collects groups of variables that are $\mathbf{0}$ at $\boldsymbol{x}$.*

*Remark* 3.9. Theorem 3.8 shows that if $\tilde{\boldsymbol{x}}_{k,t}$ falls into an $\ell_2$-ball centered at $\boldsymbol{x}^*$ with a radius no greater than $\frac{2\alpha_k\delta}{1-\epsilon+\alpha_k L}$, then `AdaHSPG+` identifies the sparsity pattern over the groups on which proximal method fails to identify, *i.e.*, $[\tilde{\boldsymbol{x}}_{k,t} + \mathbf{s}(\tilde{\boldsymbol{x}}_{k,t};\eta)]_{\mathcal{I}^0(\boldsymbol{x}^*)\cap\mathcal{I}^{\neq 0}(\tilde{\boldsymbol{x}}_{k,t})} \neq 0$. Comparing with the radius $\min\{\delta/L, \alpha_k\delta\}$ required from `Prox-SG` (Lemma A.1), `AdaHSPG+` requires an $\ell_2$-ball with larger radius, *i.e.*, $\frac{2\alpha_k\delta}{1-\epsilon+\alpha_k L} > \min\{\delta/L, \alpha_k\delta\}$ as $\alpha_k \leq 1/L$ and $\epsilon \in (0,1)$, thereby is superior to `Prox-SG` in terms of sparsity identification.

*Remark* 3.10 (`AdaHSPG+` versus `HSPG`). We provide a more detailed comparison between `AdaHSPG+` and `HSPG`. The `HSPG` algorithm is a two-stage algorithm. In the first stage, subgradient descent steps are computed for a fixed number of epochs (denoted by $K_{\text{switch}}$). The second stage uses the nonzero groups of the solution estimate computed in the first stage to define a half-space similar to, but different from, ours (more on this below). The rest of the second stage involves the repeated computation of stochastic subgradient steps followed by projections defined using the half-space. Such a design has two drawbacks. First, the switching constant $K_{\text{switch}}$ is determined offline. In practice, for each new problem, users need to spend significant time tuning this constant to get acceptable results. Second, by only performing iterations based on the half-space

after $K_{\text{switch}}$ epochs (*i.e.*, no longer allowing stochastic proximal gradient steps to be performed), it is more likely that `HSPG` will predict groups to be zero while they are in fact nonzero at the solution. This is because, in the second stage, only nonzero groups of variables are updated. To avoid this unfortunate possibility in their convergence analysis, the authors make rather strong assumptions that are either unlikely to hold in practice or impossible to verify in practice (*e.g.*, they require the point computed during the first stage to be close enough to the solution, which is typically unknown in practice). Moreover, the analysis for `HSPG` requires the regularizer to be differentiable with Lipschitz continuous gradient, which does not hold at the origin for the group regularizer considered in this paper; thus, the analysis does not apply in our setting.

To overcome the drawbacks of `HSPG`, we proposed an automatic and adaptive switching mechanism that avoids the need for a two-stage algorithm, thus avoiding the challenge of finding a good value for $K_{\text{switch}}$. To overcome the strong smoothness assumption required by `HSPG`, our `AdaHSPG+` uses a more sophisticated definition of the set $\mathcal{I}_k^{\text{HS}}$ used to compute the `Enhanced Half-Space step`. In particular, for a group $g$ to be included in $\mathcal{I}_k^{\text{HS}}$, our definition requires $[x_k]_g$ to be sufficiently far from the origin (see (3)). This is in contrast to `HSPG`, wherein *every* nonzero group is included in $\mathcal{I}_k^{\text{HS}}$. This difference, in some sense subtle, is crucial in establishing our stronger convergence theory under weaker assumptions on the regularizer. For additional aspects related to our contributions with respect to `HSPG`, see our contributions in Section 1.3.

## 4    Numerical Experiments

In this section, we demonstrate the effectiveness of `AdaHSPG+` on non-convex problems. The experimental results validate that `AdaHSPG+` significantly outperforms proximal methods in terms of significantly better group sparsity exploration and maintaining competitive objective function convergence. In Appendix **??**, we also include a convex experiments comparison.

### 4.1    Implementation details

We first describe the hyperparameters used in Algorithm 1. We designed an adaptive update for $\epsilon$. At the end of an epoch, if (i) an `Enhanced Half-Space step` was computed during the previous and current epoch, (ii) the group sparsity of the iterate did not improve compared to the iterate at the beginning of the previous epoch, and (iii) the optimality measure is sufficiently small, then we consider enlarging $\epsilon$ by setting $\epsilon \leftarrow \min\{2\epsilon, 0.999\}$. Anytime any of (i)–(iii) do not hold, we consider reducing $\epsilon$ by setting $\epsilon \leftarrow \max\{0.001, 0.5\epsilon\}$. We set $\kappa \leftarrow 10^{-4}$ to favor the inclusion of more groups from $\mathcal{G}$ into $\mathcal{I}_k^{\text{HS}}$. We set $\mu \leftarrow 1$ to give equal preference to the `Enhanced Half-Space step` and the `Prox-SVRG step`. Prior testing showed that our numerical results are rather insensitive to $\mu$ except when $\mu \gg 1$, which would significantly favor `Prox-SVRG step`. The mini-batch size $|\mathcal{B}_{k,t}|$ for all $k$ and $t$ is set to the same number $b$, whose definition will change for the convex and non-convex problem settings (these will be discussed in their corresponding sections below). We set $|\mathcal{B}_k| = N$, which is the total number of data samples. $m_h$ and $m_b$ are both set as the ratio between the total number of samples and the mini-batch size. The choice of $\eta$ and hyperparameters for `Prox-SG`, `Prox-SVRG`, and `HSPG` are discussed separately below for the convex and non-convex experiments.

### 4.2    Image classification via convolutional neural network

We now consider popular Deep Neural Networks (DNNs) for image classification tasks to demonstrate the effectiveness of `AdaHSPG+` on non-convex problems. Specifically, we use the popular benchmark DNN architectures `VGG16` (Simonyan & Zisserman, 2014), `ResNet18` (He et al., 2016), and `MobileNetV1` (Howard et al., 2017) on the two commonly tested datasets CIFAR10 (Krizhevsky & Hinton, 2009) and Fashion-MNIST (Xiao et al., 2017). As in (Li et al., 2020; Chen et al., 2021b), we conduct all experiments using 300 epochs with a mini-batch of size $b = 128$ on a GeForce RTX 2080 Ti GPU. We choose $\lambda_g = 10^{-3}$ for all $g \in \mathcal{G}$ as in Chen et al. (2021b). We remark that $\lambda_g = 10^{-3}$ is the best regularizer coefficient among the power of ten after fine-tuning with balancing the model performance and group sparsity. If $\lambda_g$ is too large, then the performance would significantly deteriorate; if $\lambda_g$ is too small, then no group sparsity is yielded. The step size $\alpha_k$ is initialized as 0.1, and then decreased by a factor of 0.1 every 75 epoch. The variables of each filter in the convolutional layer and each row of weighting matrix in the linear layer are clustered as one group following Deleu & Bengio (2021). In Table 1, we report the same metrics as

for the convex experiments, wherein all results are averaged by three runs with the best values marked as bold.

Table 1 demonstrates the superiority of `AdaHSPG+` in the non-convex setting. In particular, `AdaHSPG+` computes significantly higher group sparsity levels than all competing methods. `Prox-SVRG` does not perform well compared to the other methods since the variance reduction techniques may not work as desired for deep learning applications (Defazio & Bottou, 2019). Finally, we note that all methods perform comparably in terms of generalization error on the validation set.

Table 1: Results of non-convex problems with various neural network architectures and datasets, where we report numbers in the form of final $\psi$ / group sparsity ratio / validation accuracy for non-convex problems.

| Model | Dataset | Prox-SG | Prox-SVRG | HSPG | AdaHSPG+ |
|---|---|---|---|---|---|
| VGG16 | CIFAR10 | 0.593 / 54.0% / 90.6% | 0.825 / 14.7% / 89.4% | **0.591** / 74.6% / **91.1%** | **0.591** / **76.1%** / 91.0% |
|  | Fashion-MNIST | 0.544 / 19.1% / **93.0%** | 0.613 / 0.5% / 92.7% | 0.541 / 39.7% / **93.0%** | **0.539** / **51.2%** / 92.9% |
| ResNet18 | CIFAR10 | 0.323 / 26.5% / 94.1% | 0.361 / 2.8% / 94.2% | 0.312 / 41.6% / 94.4% | **0.311** / **42.1%** / **94.5%** |
|  | Fashion-MNIST | 0.127 / 0.0% / 94.8% | 0.145 / 0.0% / 94.6% | 0.120 / 10.4% / **94.9%** | **0.119** / **43.9%** / **94.9%** |
| MobileNetV1 | CIFAR10 | **0.401** / 58.1% / 91.7% | 0.652 / 29.2% / 90.7% | 0.403 / 65.4% / **92.0%** | 0.402 / **71.5%** / 91.8% |
|  | Fashion-MNIST | **0.229** / 62.6% / 94.2% | 0.246 / 42.0% / 94.2% | 0.230 / 74.3% / 94.5% | 0.241 / **78.9%** / **94.6%** |

*Remark* 4.1. One might ask whether we may replace `Prox-SVRG step` with vanilla `Prox-SG`? We remark that, first, `Prox-SVRG step` allows for a better theoretical convergence guarantee. Second, although `Prox-SVRG` performs worse than `Prox-SG` for deep learning tasks as *stand-alone algorithms*, `AdaHSPG+` performs better when using the `Prox-SVRG step` because of its ability to better identify the solution support; consequently, the majority of steps in `AdaHSPG+` are `Enhanced Half-Space step`s.

*Remark* 4.2. It is also interesting to compare the testing accuracy of different neural network architectures trained by sparsity-inducing algorithms and popular algorithms used in the deep learning community. Specifically, we use the stochastic gradient descent with momentum (`SGD + M`) and `AdaHSPG+` as an example, and report the results in Table 2 including two metrics. One is accuracy difference, which is defined as the prediction accuracy achieved by `AdaHSPG+` minus the prediction accuracy achieved by `SGD + M`). And the other is model compression rate. One can observe that, in most cases, for neural networks trained by `AdaHSPG+`, there is little or even no loss in the testing accuracy while the model sizes are reduced by at least 45%. This phenomenon brings benefits like faster inference and the potential deployment of models on edge devices.

Table 2: Accuracy difference and model compression rate achieved by `AdaHSPG+` and stochastic gradient descent with momentum on different neural networks and datasets.

| Dataset | Metric | | VGG16 | ResNet18 | MobileNetV1 |
|---|---|---|---|---|---|
| Cifar10 | Accuracy Difference | ($\uparrow$) | -1.6% | 1.5% | -2.6% |
|  | Model Compression Rate | ($\uparrow$) | 76.1% | 42.1% | 71.5% |
| Fashion-MINST | Accuracy Difference | ($\uparrow$) | -0.6% | 0.0% | -0.4% |
|  | Model Compression Rate | ($\uparrow$) | 51.2% | 43.9% | 78.9% |

We then investigate the evolution of our evaluation metrics during the training process. The results are shown in Figure 4 and Figure 5, which include the objective function $\psi$, group sparsity ratio, and validation accuracy. `AdaHSPG+` performs better than the other methods in terms of group sparsity since it consistently exhibits a higher group sparsity ratio in the majority of training epochs.

We next sweep the regularizer coefficient $\lambda$ over $\{10^{-2}, 10^{-3}, 10^{-4}\}$. As shown in Figure 2, `AdaHSPG+` consistently exhibits the frontiers of the Pareto curves over CIFAR10 experiments, which demonstrates the superiority of HSPG-family methods over better group sparsity exploration and competitive objective convergence in practice compared with proximal methods.

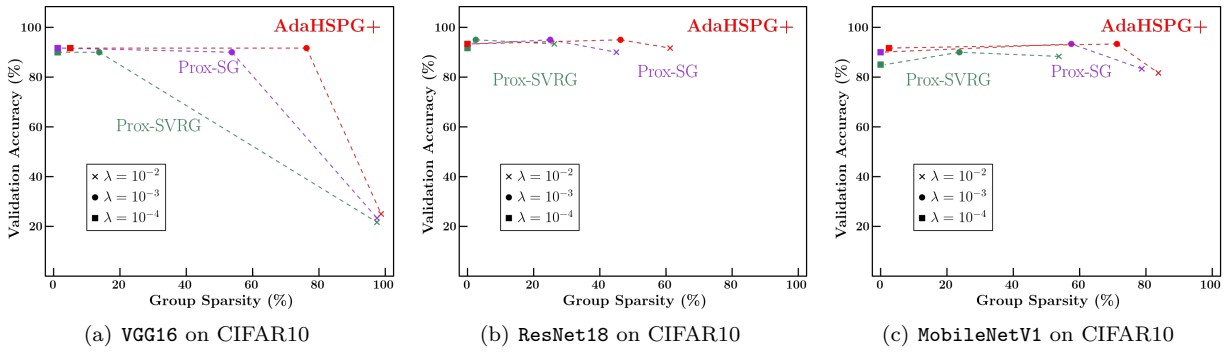

Figure 2: Validation accuracy versus group sparsity under varying regularizer coefficient $\lambda$s.

We end this section by fine-tuning the step size (learning rate) by considering multi-step (`MultiStep`) scheduler and cosine annealing scheduler (`Cosine`) (Loshchilov & Hutter, 2016) with different starting value $\alpha_0$ from $\{1.0, 0.1, 0.01\}$. For the multi-step scheduler, we divide the step size by 10 for every $\lfloor 1/4T \rfloor$, i.e., $\alpha_k = \alpha_0 \times 0.1^{\lfloor k/(0.25T) \rfloor}$, where $T$ is the total number of epochs. For

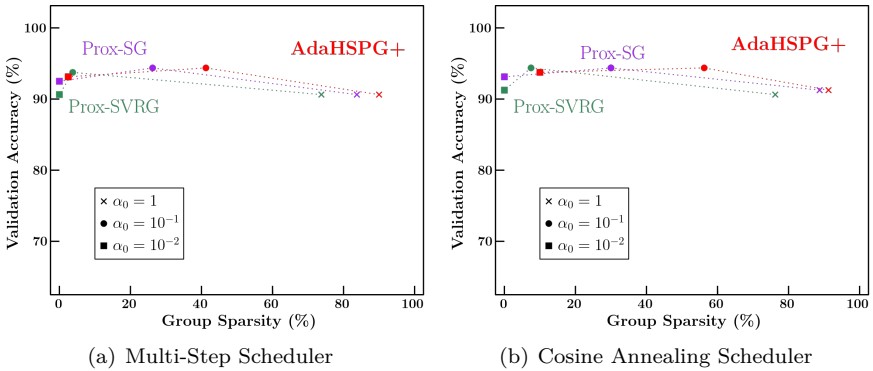

Figure 3: Validation accuracy versus group sparsity under varying step sizes.

`Cosine` scheduler, we use the default parameters provided by Pytorch with the period length as the maximum epoch $T$. The results are presented in Figure 3. We observe that for different step size schedulers and initial step sizes, `AdaHSPG+` consistently exhibits the frontier among all Patero curves. The results validate the superiority of HSPG-family methods over group sparsity exploration and achieving models with competitive generalization performance compared with proximal methods.

### 4.3 Question and answering via large-scale transformer

We follow (Chen et al., 2021b) to show the scalability of `AdaHSPG+` by training and pruning the large-scale transformer Bert (Vaswani et al., 2017), evaluated on SQuAD, a question-answering benchmark (Rajpurkar et al., 2016). The structures inside Bert consist of embedding layers, fully connected layers, and multi-head attention layers. For fairness, we do not prune the embedding layers following the prior Bert compression works (Deleu & Bengio, 2021). As (Chen et al., 2021b), we selected $\lambda_g$ as $10^{-3}$ for all groups $g \in \mathcal{G}$ with 10% and 30% group sparsity upper bound constraints. We compared with the best results of an adaptive proximal method ProxSSI presented in (Deleu & Bengio, 2021).

As shown in Table 3, `AdaHSPG+` significantly outperforms ProxSSI (Deleu & Bengio, 2021) in terms of achieving better exact match rate and F1-score and higher group sparsity levels. In particular, with a similar F1-score of around 82%, HSPG-family methods could reach 1.8x higher group sparsity (30% versus 16.5%). Furthermore, if the proximal method yielded more group sparsity from 16.5% to 23.9%, their model performance dramatically regressed to 77.5% F1-score, which was not comparable with the 82%-84% F1-scores of HSPG-family methods. The reason proximal methods are not working well for deep learning applications is that their group sparsity exploration heavily relies on the selection of the regularizer coefficient $\lambda$ and the learning rate $\alpha$. The proximal method has to increase either $\lambda$ or $\alpha$ to yield satisfactory group sparsity, which may noticeably deteriorate the model performance. In sharp contrast, HSPG-family methods feature a Half-Space projector that provides a novel mechanism for producing group sparsity which explores

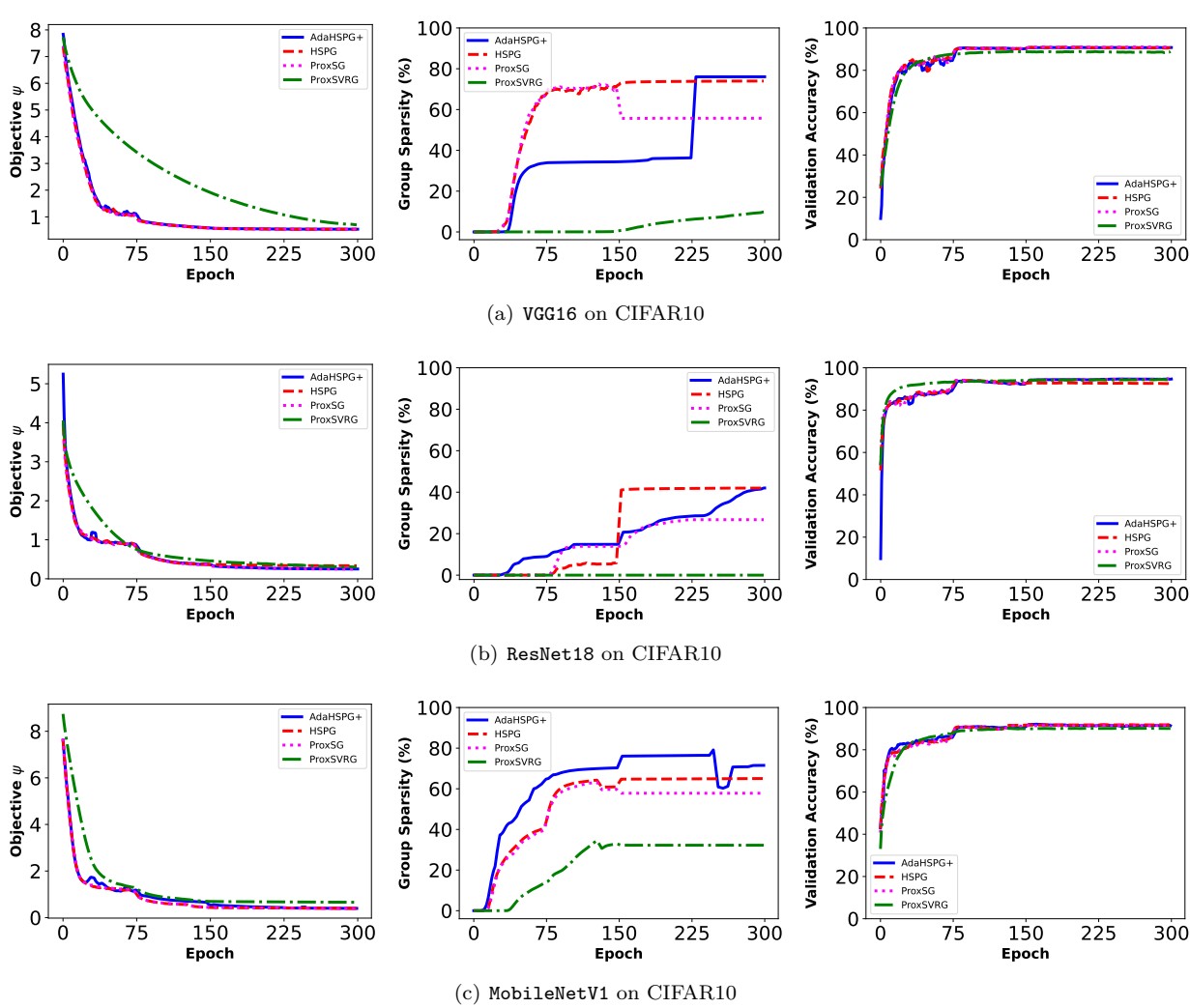

(a) `VGG16` on CIFAR10

(b) `ResNet18` on CIFAR10

(c) `MobileNetV1` on CIFAR10

Figure 4: The evolution of $\psi$, group sparsity ratio, and validation accuracy over the epochs.

Table 3: Pruning Bert on SQuAD.

| Method | Group Sparsity | Exact | F1-score |
|---|---|---|---|
| Baseline | 0% | 81.0% | 88.3% |
| ProxSSI Deleu & Bengio (2021) | 16.5% | 72.3% | 82.0% |
| ProxSSI Deleu & Bengio (2021) | 23.9% | 66.2% | 77.5% |
| HSPG Chen et al. (2021b) | 10.0% | 75.0% | 84.1% |
| HSPG Chen et al. (2021b) | 30.0% | 72.3% | 82.1% |
| `AdaHSPG+` | 10.0% | **75.2%** | **84.3%** |
| `AdaHSPG+` | 30.0% | 72.6% | 82.5% |

group sparsity without such heavy dependency over $\lambda$ and $\alpha$, thereby typically achieving solutions with competitive performance but remarkably higher group sparsity levels.

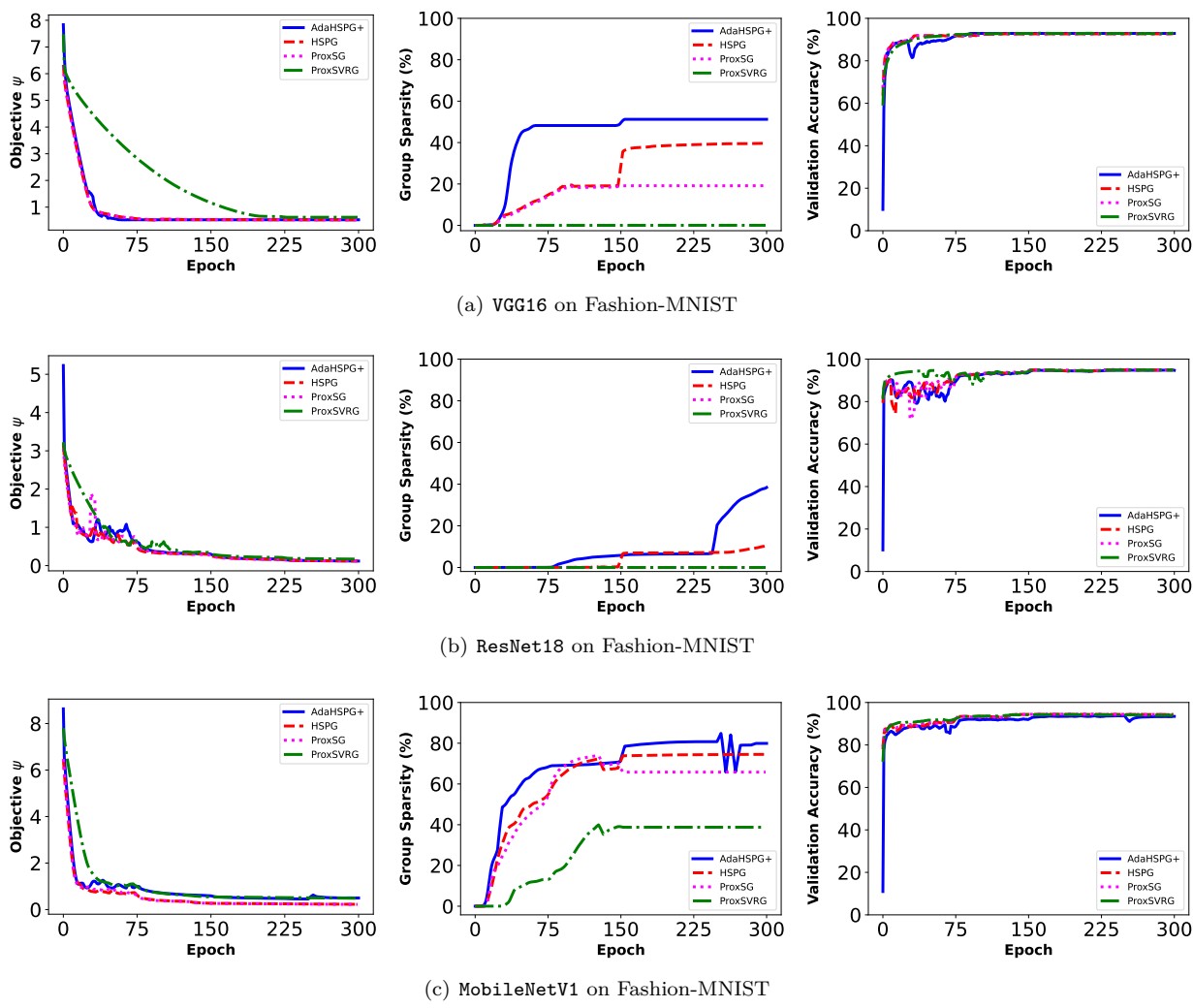

Figure 5: The evolution of $\psi$, group sparsity ratio, and validation accuracy over the epochs.

## 5 Conclusion

In this paper, we proposed `AdaHSPG+`, which is a significant upgrade of the recently proposed algorithm `HSPG`, to solve optimization problems with group sparse regularization. `AdaHSPG+` takes advantage of variance reduction techniques and is equipped with a novel adaptive strategy for estimating the support of a solution. We showed that `AdaHSPG+` has a much stronger convergence result under milder assumptions when compared with `HSPG`. Numerically, thanks to a new adaptive and automatic hyperparameter tuning strategy and switching mechanism, `AdaHSPG+` requires fewer hyper-parameter fine-tuning efforts compared to `HSPG`. Finally, the proposed `AdaHSPG+` outperforms popular stochastic proximal methods on the non-convex deep neural network benchmark problems in terms of various performance measures that include final objective function value, solution group sparsity ratio, and generalization error.

**Acknowledgments**

Guanyi Wang was supported by the NUS under AcRF Tier-1 grant A-8000607-00-00. Yutong Dai and Daniel P. Robinson were supported by the US National Science Foundation grant DMS-2012243. We are grateful for the support from Lehigh University, Microsoft, and the National University of Singapore. Finally, we thank anonymous reviewers and editors for helpful feedback.

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

## A  Proofs.

### A.1  Proof of Lemma 3.5

**Lemma 3.5.** If the `Enhanced Half-Space step` is computed during the $k$th epoch of Algorithm 1, then the $t$th iterate $\tilde{\boldsymbol{x}}_{k,t+1}$ computed in Algorithm 2 satisfies

$$
\mathbb{E}_{\mathcal{B}_{k,t}}[\psi(\tilde{\boldsymbol{x}}_{k,t+1}) \mid \tilde{\boldsymbol{x}}_{k,t}] \leq \psi(\tilde{\boldsymbol{x}}_{k,t}) - \left(\frac{1-\epsilon}{\alpha_k} - \frac{L}{2}\right)\left\|[\tilde{\boldsymbol{x}}_{k,t}]_{\mathcal{I}^{\texttt{PROJ}}_{k,t}}\right\|^2 - \alpha_k\left(1 - \frac{L\alpha_k}{2}\right)\left\|\nabla_{\mathcal{I}^{\texttt{GRAD}}_{k,t}}\psi(\tilde{\boldsymbol{x}}_{k,t})\right\|^2
$$
$$
+ \frac{\lambda_{\max}\alpha_k^2}{\kappa^2\epsilon}\sum_{g\in\mathcal{I}^{\texttt{GRAD}}_{k,t}}\|[\tilde{\boldsymbol{x}}_{k,t}]_g\| + \frac{L\alpha_k^2\sigma^2\mathbb{I}(|\mathcal{B}_{k,t}| < N)}{2|\mathcal{B}_{k,t}|},
$$

where the expectation is taken with respect to $\mathcal{B}_{k,t}$ and conditioning on $\tilde{\boldsymbol{x}}_{k,t}$, $\mathbb{I}(\cdot)$ is the indicator function,

$$
\lambda_{\max} = \max\{\lambda_g \mid g \in \mathcal{G}\},
$$
$$
\mathcal{I}^{\texttt{PROJ}}_{k,t} = \{g \in \mathcal{I}^{\texttt{HS}}_{k,t} \mid \frac{([\tilde{\boldsymbol{x}}_{k,t}]_g - \alpha_k\nabla_g\psi_{\mathcal{B}_{k,t}}(\tilde{\boldsymbol{x}}_{k,t}))^T[\tilde{\boldsymbol{x}}_{k,t}]_g}{\|[\tilde{\boldsymbol{x}}_{k,t}]_g\|^2} \leq \epsilon\},
$$
$$
\mathcal{I}^{\texttt{GRAD}}_{k,t} = \mathcal{I}^{\texttt{HS}}_{k,t} \setminus \mathcal{I}^{\texttt{PROJ}}_{k,t},
$$

and $\sigma^2$ is the bounded variance constant satisfying

$$
\mathbb{E}\left\|\nabla f_i(x) - \nabla f(x)\right\|^2 \leq \sigma^2,
$$

with the expectation taken with respect to randomly sampled $i$. (The existence of the constant $\sigma^2$ follows from the finite-sum structure of $f(x)$, the uniform sampling scheme, and Assumptions 3.1 and 3.4.)

*Proof.* We first prove the existence of $\sigma$. By Assumption 3.4 and Lipschitz continuity of $\nabla f_i(x)$, one can see that $\|\nabla f_i(x)\|$ is uniformly bounded by some constant $C$. By definition

$$
\|\nabla f(x)\|^2 = \frac{1}{N^2}\sum_{i=1}^{N}\|\nabla f_i(x)\|^2 + \frac{2}{N}\sum_{i=1}^{N-1}\sum_{j=i+1}^{N}\nabla f_i(x)^T\nabla f_j(x).
$$

Therefore,

$$
\mathbb{E}\left\|\nabla f_i(x) - \nabla f(x)\right\|^2 = \mathbb{E}[\|\nabla f_i(x)\|^2] - \|\nabla f(x)\|^2
$$
$$
= \frac{N-1}{N}\sum_{i=1}^{N}\|\nabla f_i(x)\|^2 - \frac{2}{N}\sum_{i=1}^{N-1}\sum_{j=i+1}^{N}\nabla f_i(x)^T\nabla f_j(x)
$$
$$
\leq (N-1)C^2 + \frac{2}{N}\sum_{i=1}^{N-1}\sum_{j=i+1}^{N}\|\nabla f_i(x)\|\,\|\nabla f_j(x)\|
$$
$$
\leq \sigma^2,
$$

for some $\sigma > 0$.

Next, we prove the main result. An `Enhanced Half-Space step` can be written as $\tilde{\boldsymbol{x}}_{k,t+1} = \tilde{\boldsymbol{x}}_{k,t} - \alpha_k d_{k,t}$, where

$$
[d_{k,t}]_g = \begin{cases} \nabla_g\psi_{\mathcal{B}_{k,t}}(\tilde{\boldsymbol{x}}_{k,t}) & \text{if } g \in \mathcal{I}^{\texttt{GRAD}}_{k,t} \\ [x_k]_g/\alpha_k & \text{if } g \in \mathcal{I}^{\texttt{PROJ}}_{k,t}, \\ 0 & g \in \mathcal{I}^{\texttt{NHS}}_{k,t}. \end{cases} \tag{7}
$$

Let $\xi_{k,t} \in \partial r(\tilde{\boldsymbol{x}}_{k,t})$ and $\xi_{k,t+1} \in \partial r(\tilde{\boldsymbol{x}}_{k,t+1})$. Therefore

$$
\begin{aligned}
\psi(\tilde{\boldsymbol{x}}_{k,t+1}) &= \psi(\tilde{\boldsymbol{x}}_{k,t} - \alpha_k d_{k,t}) \\
&= f(\tilde{\boldsymbol{x}}_{k,t} - \alpha_k d_{k,t}) + r(\tilde{\boldsymbol{x}}_{k,t} - \alpha_k d_{k,t}) \\
&\leq f(\tilde{\boldsymbol{x}}_{k,t}) - \alpha_k \nabla f(\tilde{\boldsymbol{x}}_{k,t})^T d_{k,t} + \frac{L}{2}\alpha_k^2 \|d_{k,t}\|^2 + r(\tilde{\boldsymbol{x}}_{k,t}) - \alpha_k \xi_{k,t+1}^T d_{k,t} \\
&= \psi(\tilde{\boldsymbol{x}}_{k,t}) - \alpha_k \nabla f(\tilde{\boldsymbol{x}}_{k,t})^T d_{k,t} + \frac{L}{2}\alpha_k^2 \|d_{k,t}\|^2 - \alpha_k \left[\xi_{k,t+1} - \xi_{k,t} + \xi_{k,t}\right]^T d_{k,t} \\
&= \psi(\tilde{\boldsymbol{x}}_{k,t}) - \alpha_k \left[\nabla f(\boldsymbol{x}_{k,t}) + \xi_{k,t}\right]^T d_{k,t} + \frac{L}{2}\alpha_k^2 \|d_{k,t}\|^2 + \alpha_k \left[\xi_{k,t} - \xi_{k,t+1}\right]^T d_{k,t},
\end{aligned}
\tag{8}
$$

where the only inequality holds by the convexity of $r$, Assumption 3.1, and (7).

Since $\xi_{k,t}$ and $\xi_{k,t+1}$ are arbitrary, consider the following specific choices:

$$
[\xi_{k,t}]_g = \begin{cases} \lambda_g \frac{[\tilde{\boldsymbol{x}}_{k,t}]_g}{\|[\tilde{\boldsymbol{x}}_{k,t}]_g\|_2}, & \text{if } g \in \mathcal{I}_{k,t}^{\neq 0} \\ \lambda_g u_g, & \text{if } g \in \mathcal{I}_{k,t}^0 \end{cases} \quad \text{and} \quad [\xi_{k,t+1}]_g = \begin{cases} \lambda_g \frac{[\tilde{\boldsymbol{x}}_{k,t+1}]_g}{\|[\tilde{\boldsymbol{x}}_{k,t+1}]_g\|}, & \text{if } g \in \mathcal{I}_{k,t+1}^{\neq 0} \\ \lambda_g \frac{[\tilde{\boldsymbol{x}}_{k,t}]_g}{\|[\tilde{\boldsymbol{x}}_{k,t}]_g\|}, & \text{if } g \in \mathcal{I}_{k,t}^{\neq 0} \cap \mathcal{I}_{k,t+1}^0 \\ \lambda_g [u]_g, & \text{if } g \in \mathcal{I}_{k,t}^0 \cap \mathcal{I}_{k,t+1}^0 \end{cases},
$$

where $\mathcal{I}_{k,t}^{\neq 0} = \{g \in \mathcal{G} \mid [\tilde{\boldsymbol{x}}_{k,t}]_g \neq 0\}$, $\mathcal{I}_{k,t}^0 = \{g \in \mathcal{G} \mid [\tilde{\boldsymbol{x}}_{k,t}]_g = 0\}$, $u_g \in \mathbb{R}^{|g|}$, and $\|u_g\| \leq 1$.

For any $g \in \mathcal{I}_{k,t}^{\texttt{PROJ}}$, by definition $g \in \mathcal{I}_{k,t}^{\neq 0}$ and $g \in \mathcal{I}_{k,t+1}^0$, it follows that

$$
[\xi_{k,t}]_g = [\xi_{k,t+1}]_g \quad \text{for all } g \in \mathcal{I}_{k,t}^{\texttt{PROJ}}.
\tag{9}
$$

If we now define $\rho_{k,t}^g = \epsilon \|[\tilde{\boldsymbol{x}}_{k,t}]_g\|$, then it follows that

$$
\begin{aligned}
\left[\xi_{k,t} - \xi_{k,t+1}\right]^T d_{k,t} &= \sum_{g \in \mathcal{I}_{k,t}^{\texttt{GRAD}}} \lambda_g [d_{k,t}]_g^T \left[\frac{[\tilde{\boldsymbol{x}}_{k,t}]_g}{\|[\tilde{\boldsymbol{x}}_{k,t}]_g\|} - \frac{[\tilde{\boldsymbol{x}}_{k,t+1}]_g}{\|[\tilde{\boldsymbol{x}}_{k,t+1}]_g\|_2}\right] \\
&\leq \sum_{g \in \mathcal{I}_{k,t}^{\texttt{GRAD}}} \lambda_g \frac{\|[d_{k,t}]_g\|}{\rho_{k,t}^g} \left\|\left[\frac{\rho_{k,t}^g [\tilde{\boldsymbol{x}}_{k,t}]_g}{\|[\tilde{\boldsymbol{x}}_{k,t}]_g\|} - \frac{\rho_{k,t}^g [\tilde{\boldsymbol{x}}_{k,t+1}]_g}{\|[\tilde{\boldsymbol{x}}_{k,t+1}]_g\|}\right]\right\| \\
&\leq \sum_{g \in \mathcal{I}_{k,t}^{\texttt{GRAD}}} \lambda_g \frac{\|[d_{k,t}]_g\|}{\rho_{k,t}^g} \|[\tilde{\boldsymbol{x}}_{k,t}]_g - [\tilde{\boldsymbol{x}}_{k,t+1}]_g\| = \sum_{g \in \mathcal{I}_{k,t}^{\texttt{GRAD}}} \lambda_g \frac{\alpha_k}{\rho_{k,t}^g} \|[d_{k,t}]_g\|^2,
\end{aligned}
$$

where the first equality holds by (9), the second inequality holds by Cauchy-Schwarz inequality, and the third inequality holds by the non-expansiveness of the Euclidean projection and the fact that $\|[\tilde{\boldsymbol{x}}_{k,t}]_g\| \geq \rho_{k,t}^g$ and $\|[\tilde{\boldsymbol{x}}_{k,t+1}]_g\| \geq \rho_{k,t}^g$ for all $g \in \mathcal{I}_{k,t}^{\texttt{GRAD}}$. We would like to point out for the third inequality that it holds as $\frac{\rho_{k,t}^g [\tilde{\boldsymbol{x}}_{k,t}]_g}{\|[\tilde{\boldsymbol{x}}_{k,t}]_g\|}$ and $\frac{\rho_{k,t}^g [\tilde{\boldsymbol{x}}_{k,t+1}]_g}{\|[\tilde{\boldsymbol{x}}_{k,t+1}]_g\|}$ are the projections of $[\tilde{\boldsymbol{x}}_{k,t}]_g$ and $[\tilde{\boldsymbol{x}}_{k,t+1}]_g$ onto the ball centered at the origin with radius $\rho_{k,t}^g$, respectively. Since $[\tilde{\boldsymbol{x}}_{k,t}]_g$ and $[\tilde{\boldsymbol{x}}_{k,t+1}]_g$ are outside the ball, for all $g \in \mathcal{I}_{k,t}^{\texttt{GRAD}}$, the result follows from the non-expansiveness of projections. Combining this inequality with (8), we obtain

$$
\begin{aligned}
\psi(\tilde{\boldsymbol{x}}_{k,t+1}) &\leq \psi(\tilde{\boldsymbol{x}}_{k,t}) - \alpha_k \left[\nabla f(\tilde{\boldsymbol{x}}_{k,t}) + \xi_{k,t}\right]^T d_{k,t} + \frac{L}{2}\alpha_k^2 \|d_{k,t}\|^2 + \alpha_k \left[\xi_{k,t} - \xi_{k,t+1}\right]^T d_{k,t} \\
&\leq \psi(\tilde{\boldsymbol{x}}_{k,t}) - \alpha_k \left[\nabla f(\tilde{\boldsymbol{x}}_{k,t}) + \xi_{k,t}\right]^T d_{k,t} + \frac{L}{2}\alpha_k^2 \|d_{k,t}\|^2 + \alpha_k^2 \sum_{g \in \mathcal{I}_{k,t}^{\texttt{GRAD}}} \lambda_g \frac{\|[d_{k,t}]_g\|^2}{\rho_{k,t}^g} \\
&= \psi(\tilde{\boldsymbol{x}}_{k,t}) - \alpha_k \left(\nabla_{\mathcal{I}_{k,t}^{\neq 0}} \psi(\tilde{\boldsymbol{x}}_{k,t})\right)^T [d_{k,t}]_{\mathcal{I}_{k,t}^{\neq 0}} + \frac{L}{2}\alpha_k^2 \left\|[d_{k,t}]_{\mathcal{I}_{k,t}^{\neq 0}}\right\|^2 + \alpha_k^2 \sum_{g \in \mathcal{I}_{k,t}^{\texttt{GRAD}}} \lambda_g \frac{\|[d_{k,t}]_g\|^2}{\rho_{k,t}^g} \\
&\leq \psi(\tilde{\boldsymbol{x}}_{k,t}) - \alpha_k \left(\nabla_{\mathcal{I}_{k,t}^{\texttt{HS}}} \psi(\tilde{\boldsymbol{x}}_{k,t})\right)^T [d_{k,t}]_{\mathcal{I}_{k,t}^{\texttt{HS}}} + \frac{L}{2}\alpha_k^2 \left\|[d_{k,t}]_{\mathcal{I}_{k,t}^{\texttt{HS}}}\right\|^2 + \frac{\lambda_{\max}\alpha_k^2}{\kappa^2 \epsilon} \sum_{g \in \mathcal{I}_{k,t}^{\texttt{GRAD}}} \|[\tilde{\boldsymbol{x}}_{k,t}]_g\|,
\end{aligned}
\tag{10}
$$

where the only equality holds from (7).

Notice that for any $g \in \mathcal{I}_{k,t}^{\text{PROJ}}$, by definition, one has $\left([\tilde{\boldsymbol{x}}_{k,t}]_g - \alpha_k \nabla_g \psi_{\mathcal{B}_{k,t}}(\tilde{\boldsymbol{x}}_{k,t})\right)^T [\tilde{\boldsymbol{x}}_{k,t}]_g \leq \epsilon \|[\tilde{\boldsymbol{x}}_{k,t}]_g\|^2$. Taking expectation with respect to $\mathcal{B}_{k,t}$, it follows that $\left([\tilde{\boldsymbol{x}}_{k,t}]_g - \alpha_k \nabla_g \psi(\tilde{\boldsymbol{x}}_{k,t})\right)^T [\tilde{\boldsymbol{x}}_{k,t}]_g \leq \epsilon \|[\tilde{\boldsymbol{x}}_{k,t}]_g\|^2$. Rearranging terms, one gets

$$-\alpha_k \left(\nabla_g \psi(\tilde{\boldsymbol{x}}_{k,t})\right)^T [\tilde{\boldsymbol{x}}_{k,t}]_g \leq (\epsilon - 1) \|[\tilde{\boldsymbol{x}}_{k,t}]_g\|^2 .$$

Summing over all $g \in \mathcal{I}_{k,t}^{\text{PROJ}}$ for the above inequality, one has

$$-\alpha_k \left(\nabla_{\mathcal{I}_{k,t}^{\text{PROJ}}} \psi(\tilde{\boldsymbol{x}}_{k,t})\right)^T [\tilde{\boldsymbol{x}}_{k,t}]_{\mathcal{I}_{k,t}^{\text{PROJ}}} \leq (\epsilon - 1) \left\|[\tilde{\boldsymbol{x}}_{k,t}]_{\mathcal{I}_{k,t}^{\text{PROJ}}}\right\|^2 . \tag{11}$$

Note that

$$\begin{aligned}
-\alpha_k \left(\nabla_{\mathcal{I}_k^{\text{HS}}} \psi(\tilde{\boldsymbol{x}}_{k,t})\right)^T [d_{k,t}]_{\mathcal{I}_k^{\text{HS}}} &= -\alpha_k \left(\nabla_{\mathcal{I}_{k,t}^{\text{GRAD}}} \psi(\tilde{\boldsymbol{x}}_{k,t})\right)^T [d_{k,t}]_{\mathcal{I}_{k,t}^{\text{GRAD}}} - \alpha_k \left(\nabla_{\mathcal{I}_{k,t}^{\text{PROJ}}} \psi(\tilde{\boldsymbol{x}}_{k,t})\right)^T [d_{k,t}]_{\mathcal{I}_{k,t}^{\text{PROJ}}} \\
&= -\alpha_k \left(\nabla_{\mathcal{I}_{k,t}^{\text{GRAD}}} \psi(\tilde{\boldsymbol{x}}_{k,t})\right)^T \left(\nabla_{\mathcal{I}_{k,t}^{\text{GRAD}}} \psi_{\mathcal{B}_{k,t}}(\tilde{\boldsymbol{x}}_{k,t})\right) - \left(\nabla_{\mathcal{I}_{k,t}^{\text{PROJ}}} \psi(\tilde{\boldsymbol{x}}_{k,t})\right)^T [\tilde{\boldsymbol{x}}_{k,t}]_{\mathcal{I}_{k,t}^{\text{PROJ}}} \\
&\leq -\alpha_k \left(\nabla_{\mathcal{I}_{k,t}^{\text{GRAD}}} \psi(\tilde{\boldsymbol{x}}_{k,t})\right)^T \left(\nabla_{\mathcal{I}_{k,t}^{\text{GRAD}}} \psi_{\mathcal{B}_{k,t}}(\tilde{\boldsymbol{x}}_{k,t})\right) - \frac{1-\epsilon}{\alpha_k} \left\|[\tilde{\boldsymbol{x}}_{k,t}]_{\mathcal{I}_{k,t}^{\text{PROJ}}}\right\|^2 ,
\end{aligned} \tag{12}$$

where the second equality holds by (7). Now, taking the expectation with respect to $\mathcal{B}_{k,t}$ on (12), one has

$$-\alpha_k \mathbb{E}\left[\left(\nabla_{\mathcal{I}_k^{\text{HS}}} \psi(\tilde{\boldsymbol{x}}_{k,t})\right)^T [d_{k,t}]_{\mathcal{I}_k^{\text{HS}}}\right] \leq -\alpha_k \left\|[\nabla \psi(\tilde{\boldsymbol{x}}_{k,t})]_{\mathcal{I}_{k,t}^{\text{GRAD}}}\right\|^2 - \frac{1-\epsilon}{\alpha_k} \left\|[\tilde{\boldsymbol{x}}_{k,t}]_{\mathcal{I}_{k,t}^{\text{PROJ}}}\right\|^2 . \tag{13}$$

Similarly,

$$\left\|[d_{k,t}]_{\mathcal{I}_k^{\text{HS}}}\right\|^2 = \left\|\nabla_{\mathcal{I}_{k,t}^{\text{GRAD}}} \psi_{\mathcal{B}_{k,t}}(\tilde{\boldsymbol{x}}_{k,t})\right\|^2 + \frac{1}{\alpha_k^2} \left\|[\tilde{\boldsymbol{x}}_{k,t}]_{\mathcal{I}_{k,t}^{\text{PROJ}}}\right\|^2 . \tag{14}$$

Since for any random vector $Y$, $\mathbb{E}[\|Y - \mathbb{E}[Y]\|^2] = \mathbb{E}[\|Y\|^2] - \|\mathbb{E}[Y]\|^2$, by taking the expectation with respect to $\mathcal{B}_{k,t}$, one obtains

$$\begin{aligned}
\mathbb{E}\left[\left\|\nabla_{\mathcal{I}_{k,t}^{\text{GRAD}}} \psi_{\mathcal{B}_{k,t}}(\tilde{\boldsymbol{x}}_{k,t})\right\|^2\right] &= \left\|\nabla_{\mathcal{I}_{k,t}^{\text{GRAD}}} \psi(\tilde{\boldsymbol{x}}_{k,t})\right\|^2 + \mathbb{E}\left[\left\|\nabla_{\mathcal{I}_{k,t}^{\text{GRAD}}} \psi_{\mathcal{B}_{k,t}}(\tilde{\boldsymbol{x}}_{k,t}) - \nabla_{\mathcal{I}_{k,t}^{\text{GRAD}}} \psi(\tilde{\boldsymbol{x}}_{k,t})\right\|^2\right] \\
&\leq \left\|\nabla_{\mathcal{I}_{k,t}^{\text{GRAD}}} \psi(\tilde{\boldsymbol{x}}_{k,t})\right\|^2 + \mathbb{E}\left[\left\|\nabla f_{\mathcal{B}_{k,t}}(\tilde{\boldsymbol{x}}_{k,t}) - \nabla f(\tilde{\boldsymbol{x}}_{k,t})\right\|^2\right] \\
&= \left\|\nabla_{\mathcal{I}_{k,t}^{\text{GRAD}}} \psi(\tilde{\boldsymbol{x}}_{k,t})\right\|^2 + \mathbb{E}\left[\left\|\frac{1}{||\mathcal{B}_{k,t}||} \sum_{i \in \mathcal{B}_{k,t}} \{\nabla f_i(\tilde{\boldsymbol{x}}_{k,t}) - \nabla f(\tilde{\boldsymbol{x}}_{k,t})\}\right\|^2\right] \\
&= \left\|\nabla_{\mathcal{I}_{k,t}^{\text{GRAD}}} \psi(\tilde{\boldsymbol{x}}_{k,t})\right\|^2 + \frac{\sigma^2 \mathbb{I}(|\mathcal{B}_{k,t}| < N)}{|\mathcal{B}_{k,t}|},
\end{aligned} \tag{15}$$

where $\mathbb{I}(\cdot)$ is the indicator function. Again, taking the expectation with respect to $\mathcal{B}_{k,t}$ on (14), together with (15), one has

$$\mathbb{E}\left[\left\|[d_{k,t}]_{\mathcal{I}_k^{\text{HS}}}\right\|^2\right] \leq \frac{\sigma^2 \mathbb{I}(|\mathcal{B}_{k,t}| < N)}{|\mathcal{B}_{k,t}|} + \left\|\nabla_{\mathcal{I}_{k,t}^{\text{GRAD}}} \psi(\tilde{\boldsymbol{x}}_{k,t})\right\|^2 + \frac{1}{\alpha_k^2} \left\|[\tilde{\boldsymbol{x}}_{k,t}]_{\mathcal{I}_{k,t}^{\text{PROJ}}}\right\|^2 . \tag{16}$$

Finally taking expectation over the entire history over (10), together with (13) and (16), one concludes

$$
\begin{aligned}
\mathbb{E}_{\mathcal{B}_{k,t}}[\psi(\tilde{\boldsymbol{x}}_{k,t+1}) \mid \tilde{\boldsymbol{x}}_{k,t}] &\leq \psi(\tilde{\boldsymbol{x}}_{k,t}) - \alpha_k \left\| [\nabla \psi(\tilde{\boldsymbol{x}}_{k,t})]_{\mathcal{I}_{k,t}^{\mathrm{GRAD}}} \right\|^2 - \frac{1-\epsilon}{\alpha_k} \left\| [\tilde{\boldsymbol{x}}_{k,t}]_{\mathcal{I}_{k,t}^{\mathrm{PROJ}}} \right\|^2 \\
&\quad + \frac{L\alpha_k^2}{2} \left[ \frac{\sigma^2 \mathbb{I}(|\mathcal{B}_{k,t}| < N)}{|\mathcal{B}_{k,t}|} + \left\| \nabla_{\mathcal{I}_{k,t}^{\mathrm{GRAD}}} \psi(\tilde{\boldsymbol{x}}_{k,t}) \right\|^2 \right] + \frac{L}{2} \left\| [\tilde{\boldsymbol{x}}_{k,t}]_{\mathcal{I}_{k,t}^{\mathrm{PROJ}}} \right\|^2 + \frac{\lambda_{\max}\alpha_k^2}{\kappa^2\epsilon} \sum_{g \in \mathcal{I}_{k,t}^{\mathrm{GRAD}}} \| [\tilde{\boldsymbol{x}}_{k,t}]_g \| \\
&= \psi(\tilde{\boldsymbol{x}}_{k,t}) - \alpha_k(1 - \frac{L\alpha_k}{2}) \left\| \nabla_{\mathcal{I}_{k,t}^{\mathrm{GRAD}}} \psi(\tilde{\boldsymbol{x}}_{k,t}) \right\|^2 - \left( \frac{1-\epsilon}{\alpha_k} - \frac{L}{2} \right) \left\| [\tilde{\boldsymbol{x}}_{k,t}]_{\mathcal{I}_{k,t}^{\mathrm{PROJ}}} \right\|^2 \\
&\quad + \frac{\lambda_{\max}\alpha_k^2}{\kappa^2\epsilon} \sum_{g \in \mathcal{I}_{k,t}^{\mathrm{GRAD}}} \| [\tilde{\boldsymbol{x}}_{k,t}]_g \| + \frac{L\alpha_k^2 \sigma^2 \mathbb{I}(|\mathcal{B}_{k,t}| < N)}{2|\mathcal{B}_{k,t}|},
\end{aligned}
$$

which completes the proof. $\qquad\square$

## A.2 Proof of Theorem 3.7

**Theorem 3.7.** Under Assumptions 3.1-3.3, and with $\epsilon \in (0,1)$, $\eta \in (0, \frac{1}{6L}]$, $\alpha_k \leq \min\left\{ \frac{2(1-\epsilon)}{L}, 1 \right\}$ and $|\mathcal{B}_k| = N$ for all $k$, $m_p(m_p - 1/2) \leq b$, and $|\mathcal{B}_{k,t}| = \min(m_p^2, N)$ for all $k$ and $t$, then Algorithm 1 generates a sequence of iterates $\{\boldsymbol{x}_k\}_{k\in\mathbb{N}}$ such that $\liminf_{k\to\infty} \mathbb{E}[\|\mathbf{s}(\boldsymbol{x}_k; \eta)\|] = 0$.

*Proof.* We first introduce the index sets

$$
\begin{align}
\mathcal{K}_P &:= \{k : k\text{th epoch proceeeds } \texttt{Prox-SVRG step}\} \quad \text{and} \tag{17} \\
\mathcal{K}_H &:= \{k : k\text{th epoch proceeeds } \texttt{Enhanced Half-Space step}\}. \tag{18}
\end{align}
$$

Summing over the first $l$ epochs and taking expectation over all past history, we have

$$
\begin{aligned}
\mathbb{E}[\psi(\boldsymbol{x}_l)] &= \mathbb{E}\left[ \psi(\boldsymbol{x}_0) + \sum_{k=1}^{l} (\psi(\boldsymbol{x}_k) - \psi(\boldsymbol{x}_{k-1})) \right] \\
&= \mathbb{E}\left[ \psi(\boldsymbol{x}_0) + \sum_{k\in\mathcal{K}_H} (\psi(\boldsymbol{x}_k) - \psi(\boldsymbol{x}_{k-1})) + \sum_{k\in\mathcal{K}_P} (\psi(\boldsymbol{x}_k) - \psi(\boldsymbol{x}_{k-1})) \right] \tag{19} \\
&\leq \psi(\boldsymbol{x}_0) - \sum_{k\in\mathcal{K}_H} \sum_{t=0}^{m_h-1} \mathbb{E}\left[ \alpha_k(1 - \frac{L\alpha_k}{2}) \left\| \nabla_{\mathcal{I}_{k,t}^{\mathrm{GRAD}}} \psi(\tilde{\boldsymbol{x}}_{k,t}) \right\|^2 + \left( \frac{1-\epsilon}{\alpha_k} - \frac{L}{2} \right) \left\| [\tilde{\boldsymbol{x}}_{k,t}]_{\mathcal{I}_{k,t}^{\mathrm{PROJ}}} \right\|^2 \right] \\
&\quad + \sum_{k\in\mathcal{K}_H} \sum_{t=0}^{m_h-1} \mathbb{E}\left[ \frac{\lambda_{\max}\alpha_k^2}{\kappa^2\epsilon} \sum_{g\in\mathcal{I}_{k,t}^{\mathrm{GRAD}}} \| [\tilde{\boldsymbol{x}}_{k,t}]_g \| + \frac{L\sigma^2\alpha_k^2}{2|\mathcal{B}_{k,t}|} \right] - \frac{1}{36L\eta^2} \sum_{k\in\mathcal{K}_P} \sum_{t=0}^{m_p-1} \mathbb{E} \left\| \mathbf{s}(\tilde{\boldsymbol{x}}_{k,t}; \eta) \right\|^2. \tag{20}
\end{aligned}
$$

(19) holds since the index set $\{1, \ldots, l\}$ can be partitioned into two disjoint sets $\mathcal{K}_H$ and $\mathcal{K}_P$. When $k \in \mathcal{K}_H$, we upper bound the $\mathbb{E}_{\{\mathcal{B}_{k,t}\}_{t=1}^{T}}[\psi(\boldsymbol{x}_k) - \psi(\boldsymbol{x}_{k-1})|\boldsymbol{x}_{k-1}]$ using Lemma 3.5; similarly we use Lemma 3.6 for $k \in \mathcal{K}_P$. Then, we use the Law of Total Expectation to get $\mathbb{E}[\psi(\boldsymbol{x}_k) - \psi(\boldsymbol{x}_{k-1})] = \mathbb{E}[\mathbb{E}_{\{\mathcal{B}_{k,t}\}_{t=1}^{T}}[\psi(\boldsymbol{x}_k) - \psi(\boldsymbol{x}_{k-1})|\boldsymbol{x}_{k-1}]]$ for the next inequality, which leads to the upper bound presented in (20). Using Assumption 3.4 yields

$$
\begin{aligned}
\mathbb{E}[\psi(\boldsymbol{x}_l)] &\leq \psi(\boldsymbol{x}_0) - \sum_{k\in\mathcal{K}_H} \sum_{t=0}^{m_h-1} \mathbb{E}\left[ \alpha_k(1 - \frac{L\alpha_k}{2}) \left\| \nabla_{\mathcal{I}_{k,t}^{\mathrm{GRAD}}} \psi(\tilde{\boldsymbol{x}}_{k,t}) \right\|^2 + \left( \frac{1-\epsilon}{\alpha_k} - \frac{L}{2} \right) \left\| [\tilde{\boldsymbol{x}}_{k,t}]_{\mathcal{I}_{k,t}^{\mathrm{PROJ}}} \right\|^2 \right] \\
&\quad + \sum_{k\in\mathcal{K}_H} \sum_{t=0}^{m_h-1} \left( \frac{\lambda_{\max}\alpha_k^2 M n_{\mathcal{G}}}{\kappa^2\epsilon} + \frac{L\sigma^2\alpha_k^2}{2b} \right) - \frac{1}{36L\eta^2} \sum_{k\in\mathcal{K}_P} \sum_{t=0}^{m_p-1} \mathbb{E} \left\| \mathbf{s}(\tilde{\boldsymbol{x}}_{k,t}; \eta) \right\|^2, \tag{21}
\end{aligned}
$$

where $n_{\mathcal{G}} = |\mathcal{G}|$. Now let $l \to \infty$, by the Assumption 3.2 and the Assumption 3.3, one concludes that

$$\sum_{k \in \mathcal{K}_H} \alpha_k \sum_{t=0}^{m_h - 1} \mathbb{E}\left[\left\|\nabla_{\mathcal{I}_{k,t}^{\text{GRAD}}} \psi(\tilde{\boldsymbol{x}}_{k,t})\right\|^2\right] < \infty, \tag{22}$$

$$\sum_{k \in \mathcal{K}_H} \sum_{t=0}^{m_h - 1} \mathbb{E}\left[\left(\frac{1-\epsilon}{\alpha_k} - \frac{L}{2}\right)\left\|[\tilde{\boldsymbol{x}}_{k,t}]_{\mathcal{I}_{k,t}^{\text{PROJ}}}\right\|^2\right] < \infty, \tag{23}$$

$$\sum_{k \in \mathcal{K}_P} \sum_{t=0}^{m_p - 1} \mathbb{E}\left\|\mathbf{s}(\tilde{\boldsymbol{x}}_{k,t}; \eta)\right\|^2 < \infty. \tag{24}$$

By the design of the algorithm, we consider three cases.

**Case-1**. $|\mathcal{K}_P| = \infty$ and $|\mathcal{K}_H| < \infty$. From (24), we conclude $\liminf_{k \to \infty} \mathbb{E}\|\mathbf{s}(\boldsymbol{x}_k; \eta)\| = 0$.

**Case-2**. $|\mathcal{K}_P| < \infty$ and $|\mathcal{K}_H| = \infty$. Since

$$\|\mathbf{s}(\boldsymbol{x}_k; \eta)\|^2 = \left\|[\mathbf{s}(\boldsymbol{x}_k; \eta)]_{\mathcal{I}_k^{\text{HS}}}\right\|^2 + \left\|[\mathbf{s}(\boldsymbol{x}_k; \eta)]_{\mathcal{I}_k^{\text{NHS}}}\right\|^2, \tag{25}$$

in order to show that $\liminf_{k \to \infty} \mathbb{E}\|\mathbf{s}(\boldsymbol{x}_k; \eta)\| = 0$, it is sufficient to show that $\liminf_{k \to \infty} \mathbb{E}\left\|[\mathbf{s}(\boldsymbol{x}_k; \eta)]_{\mathcal{I}_k^{\text{HS}}}\right\| = 0$ and $\liminf_{k \to \infty} \mathbb{E}\left\|[\mathbf{s}(\boldsymbol{x}_k; \eta)]_{\mathcal{I}_k^{\text{NHS}}}\right\| = 0$ holds simultaneously.

We first prove $\liminf_{k \to \infty} \mathbb{E}\left\|[\mathbf{s}(\boldsymbol{x}_k; \eta)]_{\mathcal{I}_k^{\text{HS}}}\right\|$. Since $|\mathcal{K}_H| = \infty$ and $\sum_k \alpha_k = \infty$, we conclude from (22) that $\liminf_{k \to \infty} \sum_{t=0}^{m_h-1} \mathbb{E}\left[\left\|\nabla_{\mathcal{I}_{k,t}^{\text{GRAD}}} \psi(\tilde{\boldsymbol{x}}_{k,t})\right\|\right] = 0$. Notice that once a group of variables being projected to 0, that group will never be included in the computation of the `Enhanced Half-Space step` anymore based on the design of `AdaHSPG+`. Hence, there exits a $K > 0$ that for all $k \geq K, t \in \{0, \dots, m_h - 1\}$, $\mathcal{I}_{k,t}^{\text{GRAD}} = \mathcal{I}_{k,t}^{\text{HS}}$ holds, which implies $\liminf_{k \to \infty} \sum_{t=0}^{m_h-1} \mathbb{E}\left[\left\|\nabla_{\mathcal{I}_{k,t}^{\text{HS}}} \psi(\tilde{\boldsymbol{x}}_{k,t})\right\|\right] = 0$. By Lemma 2.4 in Curtis et al. (2020), we have

$$\left\|\nabla_{\mathcal{I}_k^{\text{HS}}} \psi(\tilde{\boldsymbol{x}}_{k,t})\right\| \geq \left\|[\mathbf{s}(\tilde{\boldsymbol{x}}_{k,t}; \eta)]_{\mathcal{I}_k^{\text{HS}}}\right\| \tag{26}$$

holds for all $k$ and $t \in \{0, \dots, m_h - 1\}$. Since $\tilde{\boldsymbol{x}}_{k,0} = \boldsymbol{x}_k$, $\mathcal{I}_{k,0}^{\text{grad}} = \mathcal{I}_{k,t}^{\text{GRAD}}$ and (26), we have

$$\liminf_{k \to \infty} \mathbb{E}\left\|[\mathbf{s}(\boldsymbol{x}_k; \eta)]_{\mathcal{I}_k^{\text{HS}}}\right\| \leq \liminf_{k \to \infty} \sum_{t=0}^{m_h-1} \mathbb{E}\left[\left\|\nabla_{\mathcal{I}_{k,t}^{\text{HS}}} \psi(\tilde{\boldsymbol{x}}_{k,t})\right\|\right] = 0,$$

hence $\liminf_{k \to \infty} \mathbb{E}\left\|[\mathbf{s}(\boldsymbol{x}_k; \eta)]_{\mathcal{I}_k^{\text{HS}}}\right\| = 0$.

We then prove $\liminf_{k \to \infty} \mathbb{E}\left\|[\mathbf{s}(\boldsymbol{x}_k; \eta)]_{\mathcal{I}_k^{\text{NHS}}}\right\| = 0$. By the line 6 of Algorithm 1,

$$\left\|[\mathbf{s}(\boldsymbol{x}_k; \alpha_k)]_{\mathcal{I}_k^{\text{NHS}}}\right\| = \chi_k^{\text{NHS}} \leq \frac{1}{\mu} \chi_k^{\text{HS}} = \frac{1}{\mu}\left\|[\mathbf{s}(\boldsymbol{x}_k; \eta)]_{\mathcal{I}_k^{\text{HS}}}\right\| \leq \frac{1}{\mu}\left\|[\nabla \psi(\boldsymbol{x}_k)]_{\mathcal{I}_k^{\text{HS}}}\right\|, \tag{27}$$

where the final inequality holds from (26). Taking expectation and liminf on both side concludes that

$$\liminf_{k \to \infty} \mathbb{E}\left\|[\mathbf{s}(\boldsymbol{x}_k; \eta)]_{\mathcal{I}_k^{\text{NHS}}}\right\| \leq \frac{1}{\mu} \liminf_{k \to \infty} \mathbb{E}\left\|[\nabla \psi(\boldsymbol{x}_k)]_{\mathcal{I}_k^{\text{HS}}}\right\| = 0. \tag{28}$$

**Case-3**. $|\mathcal{K}_P| = \infty$ and $|\mathcal{K}_H| = \infty$, then the result follows from the Case 1.

$\square$

## A.3   Proof of Theorem 3.8

In this section, we compare the group sparsity identification property of `AdaHSPG+` and `Prox-SG`. First, recall the non-degeneracy assumption

$$0 < \delta := \frac{1}{2} \min_{g \in \mathcal{I}^0(\boldsymbol{x}^*)} \inf_{\mathcal{B} \subseteq [n]} \left( \lambda - \|[\nabla f_{\mathcal{B}}(\boldsymbol{x}^*)]_g\| \right). \tag{29}$$

**Theorem 3.8**   Given any $k \in \mathbb{N}_+$, if the $k$th epoch performs an `Enhanced Half-Space step` with $\alpha_k \le 1/L$, then for any $t \in \{0, \dots, m_h - 1\}$ satisfying $\|\tilde{\boldsymbol{x}}_{x,t} - \boldsymbol{x}^*\| \le \frac{2\alpha_k \delta}{1 - \epsilon + \alpha_k L}$, $[\tilde{\boldsymbol{x}}_{k,t} + \mathbf{s}(\tilde{\boldsymbol{x}}_{k,t}; \eta)]_{\mathcal{I}^0(\boldsymbol{x}^*) \cap \mathcal{I}^{\neq 0}(\tilde{\boldsymbol{x}}_{k,t})} \ne 0$, and $\kappa \le \min_{g \in \mathcal{I}^0(\boldsymbol{x}^*)} \|[\tilde{\boldsymbol{x}}_{k,t}]_g\| / \|[\nabla \psi_{\mathcal{B}_{k,t}}(\tilde{\boldsymbol{x}}_{k,t})]_g\|$, it holds that `AdaHSPG+` yields $\mathcal{I}^0(\boldsymbol{x}^*) \subseteq \mathcal{I}^0(\tilde{\boldsymbol{x}}_{k,t+1})$, where $\mathcal{I}^0(\boldsymbol{x})$ collects groups of variables that are $\mathbf{0}$ at $\boldsymbol{x}$.

*Proof.* First, if $g \in \mathcal{I}^0(\boldsymbol{x}^*) \bigcap \mathcal{I}^0(\tilde{\boldsymbol{x}}_{k,t})$, then we have $[\tilde{\boldsymbol{x}}_{k,t}]_g = 0$, which implies $g \in \mathcal{I}^{\text{NHS}}_{k,t}$. Thus $[\tilde{\boldsymbol{x}}_{k,t}]_g = [\tilde{\boldsymbol{x}}_{k,t+1}]_g = 0$ as the $k$th epoch perform `Enhanced Half-Space step`, and therefore $g \in \mathcal{I}^0(\boldsymbol{x}^*) \bigcap \mathcal{I}^0(\tilde{\boldsymbol{x}}_{k,t+1})$.

Now, if $g \in \mathcal{I}^0(\boldsymbol{x}^*) \bigcap \mathcal{I}^{\neq 0}(\tilde{\boldsymbol{x}}_{k,t})$, for any inner-iteration $t \in \{0, \dots, m_h - 1\}$, there are three cases to consider.

- **Case-1.** $g \in \mathcal{I}^0(\boldsymbol{x}^*) \cap \mathcal{I}^{\text{HS}}_{k,t}$.

- **Case-2.** $g \in \mathcal{I}^0(\boldsymbol{x}^*) \cap \mathcal{I}^{\neq 0}(\tilde{\boldsymbol{x}}_{k,t}) \cap \{g \in \mathcal{G} : [\tilde{\boldsymbol{x}}_{k,t}]_g + [\mathbf{s}(\tilde{\boldsymbol{x}}_{k,t}; \eta)]_g = 0\}$.

- **Case-3.** $g \in \mathcal{I}^0(\boldsymbol{x}^*) \cap \mathcal{I}^{\neq 0}(\tilde{\boldsymbol{x}}_{k,t}) \cap \{g \in \mathcal{G} : \|[\tilde{\boldsymbol{x}}_{k,t}]_g\| < \kappa \|\nabla_g \psi_{\mathcal{B}_{k,t}}(\tilde{\boldsymbol{x}}_{k,t})\|\}$.

Based on the conditions provided above for Theorem 3.8, the **Case-2** does not hold due to condition $[\tilde{\boldsymbol{x}}_{k,t}]_g + [\mathbf{s}(\tilde{\boldsymbol{x}}_{k,t}; \eta)]_g \ne 0$, and the **Case-3** does not hold due to condition $\kappa \le \min_{g \in \mathcal{I}^0(\boldsymbol{x}^*)} \|[\tilde{\boldsymbol{x}}_{k,t}]_g\| / \|[\nabla \psi_{\mathcal{B}_{k,t}}(\tilde{\boldsymbol{x}}_{k,t})]_g\|$. Therefore, it is sufficient to consider $g \in \mathcal{I}^0(\boldsymbol{x}^*) \cap \mathcal{I}^{\text{HS}}_{k,t}$.

$$
\begin{aligned}
& [\tilde{\boldsymbol{x}}_{k,t} - \alpha_k \nabla \psi_{\mathcal{B}_{k,t}}(\tilde{\boldsymbol{x}}_{k,t})]_g^T [\tilde{\boldsymbol{x}}_{k,t}]_g - \epsilon \|[\tilde{\boldsymbol{x}}_{k,t}]_g\|^2 \\
=& \|[\tilde{\boldsymbol{x}}_{k,t}]_g\|^2 - \alpha_k [\nabla \psi_{\mathcal{B}_{k,t}}(\tilde{\boldsymbol{x}}_{k,t})]_g^T [\tilde{\boldsymbol{x}}_{k,t}]_g - \epsilon \|[\tilde{\boldsymbol{x}}_{k,t}]_g\|^2 \\
=& (1 - \epsilon) \|[\tilde{\boldsymbol{x}}_{k,t}]_g\|^2 - \alpha_k \left( [\nabla f_{\mathcal{B}_{k,t}}(\tilde{\boldsymbol{x}}_{k,t})]_g + \lambda \frac{[\tilde{\boldsymbol{x}}_{k,t}]_g}{\|[\tilde{\boldsymbol{x}}_{k,t}]_g\|} \right)^T [\tilde{\boldsymbol{x}}_{k,t}]_g \\
=& (1 - \epsilon) \|[\tilde{\boldsymbol{x}}_{k,t}]_g\|^2 - \alpha_k [\nabla f_{\mathcal{B}_{k,t}}(\tilde{\boldsymbol{x}}_{k,t})]_g^T [\tilde{\boldsymbol{x}}_{k,t}]_g - \alpha_k \lambda \|[\tilde{\boldsymbol{x}}_{k,t}]_g\| \\
\le& (1 - \epsilon) \|[\tilde{\boldsymbol{x}}_{k,t}]_g\|^2 + \alpha_k \|[\nabla f_{\mathcal{B}_{k,t}}(\tilde{\boldsymbol{x}}_{k,t})]_g\| \|[\tilde{\boldsymbol{x}}_{k,t}]_g\| - \alpha_k \lambda \|[\tilde{\boldsymbol{x}}_{k,t}]_g\| \\
=& \|[\tilde{\boldsymbol{x}}_{k,t}]_g\| \left\{ (1 - \epsilon) \|[\tilde{\boldsymbol{x}}_{k,t}]_g\| + \alpha_k \|[\nabla f_{\mathcal{B}_{k,t}}(\tilde{\boldsymbol{x}}_{k,t})]_g\| - \alpha_k \lambda \right\}
\end{aligned}
\tag{30}
$$

By the Lipschitz continuity of $\nabla f$, we have that for each $g \in \mathcal{I}^0(\boldsymbol{x}^*) \bigcap \mathcal{I}^{\neq 0}(\tilde{\boldsymbol{x}}_{k,t})$,

$$
\begin{aligned}
\left\| [\nabla f_{\mathcal{B}_{k,t}}(\tilde{\boldsymbol{x}}_{k,t}) - \nabla f_{\mathcal{B}_{k,t}}(\boldsymbol{x}^*)]_g \right\| &\le L \|[\tilde{\boldsymbol{x}}_{k,t} - \boldsymbol{x}^*]_g\| = L \|[\tilde{\boldsymbol{x}}_{k,t}]_g\| \\
\left\| [\nabla f_{\mathcal{B}_{k,t}}(\tilde{\boldsymbol{x}}_{k,t})]_g \right\| &\le L \|[\tilde{\boldsymbol{x}}_{k,t}]_g\| + \left\| [\nabla f_{\mathcal{B}_{k,t}}(\boldsymbol{x}^*)]_g \right\|.
\end{aligned}
\tag{31}
$$

Combining with the definition of $\delta$, which implies that $\left\| [\nabla f_{\mathcal{B}_{k,t}}(\boldsymbol{x}^*)]_g \right\| \le \lambda - 2\delta$ that

$$\left\| [\nabla f_{\mathcal{B}_{k,t}}(\tilde{\boldsymbol{x}}_{k,t})]_g \right\| \le L \|[\tilde{\boldsymbol{x}}_{k,t}]_g\| + \lambda - 2\delta. \tag{32}$$

Hence combining with $\|[\tilde{\boldsymbol{x}}_{k,t}]_g\| \le \frac{2\alpha_k \delta}{1 - \epsilon + \alpha_k L}$, (30) can be further written as

$$
\begin{aligned}
& [\tilde{\boldsymbol{x}}_{k,t} - \alpha_k \nabla \psi_{\mathcal{B}_{k,t}}(\tilde{\boldsymbol{x}}_{k,t})]_g^T [\tilde{\boldsymbol{x}}_{k,t}]_g - \epsilon \|[\tilde{\boldsymbol{x}}_{k,t}]_g\|^2 \\
\le& \|[\tilde{\boldsymbol{x}}_{k,t}]_g\| \left\{ (1 - \epsilon) \|[\tilde{\boldsymbol{x}}_{k,t}]_g\| + \alpha_k \|[\nabla f_{\mathcal{B}_{k,t}}(\tilde{\boldsymbol{x}}_{k,t})]_g\| - \alpha_k \lambda \right\} \\
\le& \|[\tilde{\boldsymbol{x}}_{k,t}]_g\| \left\{ (1 - \epsilon) \|[\tilde{\boldsymbol{x}}_{k,t}]_g\| + \alpha_k L \|[\tilde{\boldsymbol{x}}_{k,t}]_g\| + \alpha_k \lambda - 2\alpha_k \delta - \alpha_k \lambda \right\} \\
=& \|[\tilde{\boldsymbol{x}}_{k,t}]_g\| \left\{ (1 - \epsilon + \alpha_k L) \|[\tilde{\boldsymbol{x}}_{k,t}]_g\| - 2\alpha_k \delta \right\} \\
\le& \|[\tilde{\boldsymbol{x}}_{k,t}]_g\| \left\{ (1 - \epsilon + \alpha_k L) \frac{2\alpha_k \delta}{1 - \epsilon + \alpha_k L} - 2\alpha_k \delta \right\} \\
=& \|[\tilde{\boldsymbol{x}}_{k,t}]_g\| (2\alpha_k \delta - 2\alpha_k \delta) = 0,
\end{aligned}
\tag{33}
$$

which shows that $[\tilde{\boldsymbol{x}}_{k,t} - \alpha_k \nabla \psi_{\mathcal{B}_{k,t}}(\tilde{\boldsymbol{x}}_{k,t})]_g^T [\tilde{\boldsymbol{x}}_{k,t}]_g \leq \epsilon \|[\tilde{\boldsymbol{x}}_{k,t}]_g\|^2$. Hence the group projection operator is triggered on $g$ to map the variables to zero, then $g \in \mathcal{I}^0(\tilde{\boldsymbol{x}}_{k,t+1})$, i.e., $[\tilde{\boldsymbol{x}}_{k,t+1}]_g = 0$. Therefore, the group sparsity of $\boldsymbol{x}^*$ can be successfully identified by `Half-Space step`, i.e., $\mathcal{I}^0(\boldsymbol{x}^*) \subseteq \mathcal{I}^0(\tilde{\boldsymbol{x}}_{k,t+1})$. $\qquad\square$

We show the generic sparsity identification property of `Prox-SG` for any mixed $\ell_1/\ell_p$ regularization for $p \geq 1$.

**Lemma A.1.** *If* $\|\boldsymbol{x}_k - \boldsymbol{x}^*\|_p \leq \min\{\delta/L, \alpha_k \delta\}$, *where* $1/p + 1/p' = 1$ ($p' = \infty$ *if* $p = 1$), *then the* `Prox-SG` *yields that for each* $g \in \mathcal{I}^0(\boldsymbol{x}^*)$, $[\boldsymbol{x}_{k+1}]_g = 0$ *holds, i.e.,* $\mathcal{I}^0(\boldsymbol{x}^*) \subseteq \mathcal{I}^0(\boldsymbol{x}_{k+1})$.

*Proof.* It follows from the reverse triangle inequality, basic norm inequalities, Lipschitz continuity of $\nabla f(\boldsymbol{x})$ and the assumption of this lemma that for any $g \in \mathcal{G}$,

$$
\begin{aligned}
\|[\nabla f_{\mathcal{B}_k}(\boldsymbol{x}_k)]_g\|_{p'} - \|[\nabla f_{\mathcal{B}_k}(\boldsymbol{x}^*)]_g\|_{p'} &\leq \|[\nabla f_{\mathcal{B}_k}(\boldsymbol{x}_k) - \nabla f_{\mathcal{B}_k}(\boldsymbol{x}^*)]_g\|_{p'} \\
&\leq \|\nabla f_{\mathcal{B}_k}(\boldsymbol{x}_k) - \nabla f_{\mathcal{B}_k}(\boldsymbol{x}^*)\|_{p'} \\
&\leq L \|\boldsymbol{x}_k - \boldsymbol{x}^*\|_p \leq L \cdot \frac{\delta}{L} = \delta.
\end{aligned}
\tag{34}
$$

By (34), we have that for any $g \in \mathcal{I}^0(\boldsymbol{x}^*)$,

$$
\|[\nabla f_{\mathcal{B}_k}(\boldsymbol{x}_k)]_g\|_{p'} \leq \|[\nabla f_{\mathcal{B}_k}(\boldsymbol{x}^*)]_g\|_{p'} + \delta \leq \lambda - 2\delta + \delta = \lambda - \delta,
\tag{35}
$$

where the final inequality holds due to $\|[\nabla f_{\mathcal{B}_k}(\boldsymbol{x}^*)]_g\|_{p'} \leq \lambda - 2\delta$ by the definition of $\delta$ proposed before. Combining (35) and the assumption of this lemma, the following holds for any $\alpha_k > 0$ that

$$
\|[\boldsymbol{x}_k - \alpha_k \nabla f_{\mathcal{B}_k}(\boldsymbol{x}_k)]_g\|_{p'} \leq \|[\boldsymbol{x}_k]_g\|_{p'} + \|[\alpha_k \nabla f_{\mathcal{B}_k}(\boldsymbol{x}_k)]_g\|_{p'} \leq \alpha_k \delta + \alpha_k(\lambda - \delta) = \alpha_k \lambda
\tag{36}
$$

which further implies that the Euclidean projection yields that

$$
\text{Proj}_{\mathcal{B}(\|\cdot\|_p, \alpha_k \lambda)}^{\text{Euclidean}}([\boldsymbol{x}_k - \alpha_k \nabla f_{\mathcal{B}_k}(\boldsymbol{x}_k)]_g) = [\boldsymbol{x}_k - \alpha_k \nabla f_{\mathcal{B}_k}(\boldsymbol{x}_k)]_g.
\tag{37}
$$

Combining with (37), the fact that proximal operator is the residual of identity operator subtracted by Euclidean project operator onto the dual norm ball and $[\boldsymbol{x}_k]_g = 0$ for any $g \in \mathcal{I}^0(\boldsymbol{x}^*)$ (Chen, 2018), we have

$$
\begin{aligned}
[\boldsymbol{x}_{k+1}]_g &= \mathbf{Prox}_{\alpha_k \lambda \|\cdot\|_p} ([\boldsymbol{x}_k - \alpha_k \nabla f_{\mathcal{B}_k}(\boldsymbol{x}_k)]_g) \\
&= \left[ I - \text{Proj}_{\mathcal{B}(\|\cdot\|_p, \alpha_k \lambda)}^{\text{Euclidean}} \right] [\boldsymbol{x}_k - \alpha_k \nabla f_{\mathcal{B}_k}(\boldsymbol{x}_k)]_g \\
&= [\boldsymbol{x}_k - \alpha_k \nabla f_{\mathcal{B}_k}(\boldsymbol{x}_k)]_g - [\boldsymbol{x}_k - \alpha_k \nabla f_{\mathcal{B}_k}(\boldsymbol{x}_k)]_g = 0,
\end{aligned}
\tag{38}
$$

consequently $\mathcal{I}^0(\boldsymbol{x}^*) \in \mathcal{I}^0(\boldsymbol{x}_{k+1})$, which completes the proof. $\qquad\square$

# B  Additional Experiments

## B.1  Convex setting

As in (Xiao & Zhang, 2014; Curtis et al., 2020), we first consider the convex logistic regression problem with mixed $\ell_1/\ell_2$-regularization for binary classification. Given $N$ data samples $(\boldsymbol{d}_1, l_1), \ldots, (\boldsymbol{d}_N, l_N)$, where $\boldsymbol{d}_i \in \mathbb{R}^n$ and $l_i \in \{-1, 1\}$ denote the feature vector of the $i$th data sample and the corresponding ground truth label, respectively, the problem is formulated as

$$
\min_{(\boldsymbol{x}; b) \in \mathbb{R}^{n+1}} \frac{1}{N} \sum_{i=1}^{N} \log(1 + e^{-l_i(\boldsymbol{x}^T \boldsymbol{d}_i + b)}) + \sum_{g \in \mathcal{G}} \lambda_g \|[\boldsymbol{x}]_g\|
$$

where $b \in \mathbb{R}$ is the bias. We consider different regularization strengths by choosing $\lambda_g$ from $\{10^{-2}, 10^{-3}, 10^{-4}\}$ for all $g \in \mathcal{G}$ and lead to different levels of sparsity in the obtained solutions, correspondingly. To form $\mathcal{G}$, we sequentially go through the variables collecting them into 100 groups.

We select all datasets from the LIBSVM repository (Chang & Lin, 2011) that (i) have at least 100 features so that the formulation of $\mathcal{G}$ is well defined, (ii) have at least $10^4$ samples so that a stochastic approach is appropriate, (iii) can be stored in main memory (16 GB), and (iv) have an accompanying test set so that the testing error can be computed to evaluate the models obtained. Based on these criteria, we ended up with the four publicly available large-scale datasets a9a, kdda, rcv1, and w8a (see Table 4 for more details). All convex experiments are conducted on a 64-bit operating system with an Intel i7-7700K CPU @ 4.20 GHz and 16 GB random-access memory.

Table 4: Summary of datasets.

| Dataset | # of Training Samples | # of Testing Samples | # of Features | Attribute |
|---------|----------------------|----------------------|---------------|-----------|
| a9a | $3.26 \times 10^4$ | $1.63 \times 10^4$ | $1.23 \times 10^2$ | binary $\{0, 1\}$ |
| kdda | $8.41 \times 10^6$ | $5.10 \times 10^6$ | $2.02 \times 10^7$ | real $[-1, 4]$ |
| rcv1 | $2.02 \times 10^4$ | $6.77 \times 10^6$ | $4.72 \times 10^4$ | unit-length |
| w8a | $4.97 \times 10^4$ | $1.50 \times 10^4$ | $3.00 \times 10^2$ | binary $\{0, 1\}$ |

All algorithms tested were run for a maximum of 100 epochs. The mini-batch size $b$ is set to $\min\{256, \lceil 0.01N \rceil\}$ as suggested in (Yang et al., 2019). The step size $\alpha_k$ is set following the approach in (Xiao & Zhang, 2014), wherein an upper bound for the Lipschitz constant $L$ is computed as $\max_i \|\boldsymbol{d}_i\|^2/4$, and then $\alpha_k \equiv 4/\max_i \|\boldsymbol{d}_i\|^2$ for `Prox-SG`, `Prox-SVRG`, `HSPG`, and `AdaHSPG+`. For `HSPG` we set $\epsilon \leftarrow 0$ and switch to the `Half-Space step` after 50 epochs (these are the default values for `HSPG`).

We use the final objective value of $\psi$, group sparsity ratio, and testing accuracy as metrics to evaluate the performance of an algorithm. The group sparsity ratio is the ratio between the number of zero groups in the solution returned and the total number of groups. All metrics are obtained by averaging across five independent runs. We report the results for $\lambda_g \in \{10^{-2}, 10^{-3}, 10^{-4}\}$ for all $g \in \mathcal{G}$ in Table 5, where we mark the best values in bold font to facilitate comparison. As shown in the tables, larger values for $\lambda_g$ result in greater group sparsity ratios and worse final objective values and testing accuracy. One also can observe that the solutions with $\lambda_g = 10^{-2}$ are zeros for the dataset rcv1 and the solutions with $\lambda_g = 10^{-3}$ and $\lambda_g = 10^{-3}$ are full dense for some datasets. This is indeed expected behavior since, for a given problem, there are two thresholds $\bar{\lambda} > 0$ and $\underline{\lambda} > 0$. Whenever $\lambda_g \geq \bar{\lambda}$, the solution becomes 0, i.e., the group sparse ratio is 100%; on the other hand, when $\lambda_g \leq \underline{\lambda}$, the solution becomes fully dense, i.e., group sparse ratio is 0%. $\bar{\lambda}$ and $\underline{\lambda}$ varies from one problem to the other.

`AdaHSPG+` and `HSPG` clearly perform the best in terms of the group sparsity ratio over the test problems, and all of the algorithms perform similarly in terms of the objective function value and testing accuracy.

Table 5: Results convex problems for $\lambda_g \in \{10^{-2}, 10^{-3}, 10^{-4}\}$ for all $g \in \mathcal{G}$, where we report numbers in the form of final $\psi$ / group sparsity ratio / testing accuracy.

| Dataset | Prox-SG | Prox-SVRG | HSPG | AdaHSPG+ |
|---------|---------|-----------|------|----------|
| | | $\lambda_g = 10^{-2}$ | | |
| a9a | **0.438** / 73.0% / 83.6% | **0.438** / **86.0%** / **83.7%** | **0.438** / **86.0%** / **83.7%** | **0.438** / **86.0%** / **83.7%** |
| kdda | **0.183** / 98.0% / **87.1%** | **0.183** / **99.0%** / **87.1%** | 0.184 / **99.0%** / 86.9% | **0.183** / **99.0%** / **87.1%** |
| rcv1 | **0.693** / **100.0%** / **47.5%** | **0.693** / **100.0%** / **47.5%** | **0.693** / **100.0%** / **47.5%** | **0.693** / **100.0%** / **47.5%** |
| w8a | 0.167 / 78.0% / **97.0%** | 0.166 / **99.0%** / **97.0%** | 0.166 / **99.0%** / **97.0%** | 0.166 / **99.0%** / **97.0%** |
| | | $\lambda_g = 10^{-3}$ | | |
| a9a | **0.347** / 28.0% / **85.0%** | **0.347** / 58.0% / **85.0%** | **0.347** / **60.0%** / **85.0%** | **0.347** / 58.0% / **85.0%** |
| kdda | **0.133** / **0.0%** / **89.3%** | **0.133** / **0.0%** / 89.2% | **0.133** / **0.0%** / **89.3%** | **0.133** / **0.0%** / **89.3%** |
| rcv1 | **0.461** / 0.0% / **93.2%** | 0.464 / 44.2% / **93.2%** | **0.461** / **47.0%** / **93.2%** | **0.461** / **47.0%** / **93.2%** |
| w8a | 0.118 / 3.0% / **97.2%** | 0.119 / 16.5% / **97.2%** | **0.113** / **36.0%** / **97.2%** | **0.113** / **36.0%** / **97.2%** |
| | | $\lambda_g = 10^{-4}$ | | |
| a9a | **0.327** / 3.0% / **85.0%** | **0.327** / 15.0% / **85.0%** | **0.327** / **25.0%** / **85.0%** | **0.327** / **25.0%** / **85.0%** |
| kdda | **0.112** / **0.0%** / **89.5%** | **0.112** / **0.0%** / **89.5%** | **0.112** / **0.0%** / **89.5%** | **0.112** / **0.0%** / **89.5%** |
| rcv1 | **0.164** / **0.0%** / **95.9%** | 0.176 / **0.0%** / 95.8% | **0.164** / **0.0%** / **95.9%** | **0.164** / **0.0%** / **95.9%** |
| w8a | 0.097 / **0.0%** / **97.5%** | **0.096** / **0.0%** / **97.5%** | 0.097 / **0.0%** / **97.5%** | **0.096** / **0.0%** / **97.5%** |

