# OpenReview forum: "An Adaptive Half-Space Projection Method for Stochastic Optimization Problems with Group Sparse Regularization"
_TMLR — Accepted by TMLR_

### Review · Reviewer_Xfq1 · 2023-03-24

**Summary Of Contributions:**

The authors proposed an Adaptive Half-Space Projection Method for composite optimization. In the paper, authors analyzed the convergence of the proposed algorithm and proved that it can converge to a stable point finally. The authors also compared the proposed method with related methods on both convex problems and non-convex problems. Results show that it converged to a better sparse solution than compared methods.

**Audience:**

Yes

**Claims And Evidence:**

Yes

**Requested Changes:**

1. A more compressive theoretical analysis is required. The authors should show the convergence rate of the proposed method at least and converging faster than compared methods.

2. For a fair comparison, a sweep of hyper-parameter for all methods are required. More details about how these values are selected should be clear in the paper.

3. If considering the convex setup in the experiment, it is better to provide theoretical analysis in the paper too. A faster convergence rate should be achieved compared to the non-convex setup. Otherwise, it is better to focus on the non-convex setup and perform more experiments. From the existing figures, it is hard to believe that the proposed method outperform other methods consistently.

**Strengths And Weaknesses:**

Strengths:
1. The authors analyzed the convergence of the proposed method.
2. The authors also compared the proposed method with related methods on both convex problems and non-convex problems. Results show that it converged to a better sparse solution than compared methods.

Weaknesses
1. It is easy to combine multiple tricks and apply it for convex or non-convex optimization. Therefore, it is critical to provide the theoretical convergence rate showing that the proposed method does converges faster than compared methods. However, this paper only proved convergence and it is weak.
2. The experimental comparison is also a bit weak. Since the authors did not show any convergence analysis for the convex problem, it is better to focus on the non-convex problem only.  The improvement in Table 1 is also not significant.
3. The  hyper-parameter details in the experiments for compared methods are not clear and using default value is not fair either. For a fair comparison, a sweep of hyper-parameter for all methods are required.

---

> ### Comment · Reviewer_jnWn · 2023-04-14
> **Agree with concerns**
>
> Reviwer Xfq1 makes an interesting point that perhaps if the convex setting was omitted from the paper entirely (focusing on the non-convex) it may be improved. For example if some/all of the downfalls of HSPG were explicitly shown to be improved (like overcoming the permanent zeroing-out of groups). And I agree with the comments about hyperparameter sweeping, and the insignificance of Table 1 (as described in my review)

---

> ### Author Response · Authors · 2023-04-19
> **Revisions and more experiments are available.**
>
> Dear Reviewer Xfq1,
>
> We appreciate your constructive comments and valued suggestions. We follow your sugestions to conduct more non-convex experiments and hyper-parameter tuning. Please see our responses as follows. Look forward to further discussion.
>
> - **Lack of convergence rate.**
>
> > Thank you for the question. Please see our general response above where we hope that we adequately addressed your concern.
>
> - **More experiments if no convergence rate analysis.**
>
> > Thanks for the suggestion. We have conducted and included one more non-convex experiment on large-scale transformer, Bert on SQuAD, in the revision. We are studying a super-resolution via CARN network, which will be included into the final version. For Bert on SQuAD, we compared with an adaptive proximal method ProxSSI, the results of which further demonstrate the superiority of the proposed AdaHSPG+ optimizer over proximal methods in terms of effectively producing group sparsity and maintaining competitive objective convergence in practice.
>
> - **A sweep over the hyper-parameters is needed.**
>
> > Thanks for the suggestion. Please refer to our general response above for more details. We have included comprehensive ablation studies regarding the two main hyper-parameters of proximal methods, i.e., the learning rate and regularizer coefficient. The results show that AdaHSPG+ consistently outperform proximal methods in terms of group sparsity while maintaining a competitive objective function value.
>
> Sincerely,
>
> Paper 837 Authors

---

### Review · Reviewer_riu1 · 2023-03-28

**Summary Of Contributions:**

This paper proposes a new method (AdaHSPG+) for nonconvex stochastic optimization problems under the group-sparse regularization (i.e., mixed $\ell_1/\ell_2$ problem). AdaHSPG+ resolves the several limitations of the most relevant method (HSPG) summarized below:

(i) HSPG is a two-stage method where the second stage requires a sufficiently accurate solution provided by the first stage to achieve better performance, resulting in the costly tuning of switching time.

(ii) The convergence of HSPG is not guaranteed for mixed $\ell_1/\ell_2$ problem because it is not Lipschitz continuous in a neighborhood of the origin.

(iii) Limited empirical performance due to the expensive tuning cost of crucial hyperparameters.

The proposed method (AdaHSPG+) is not a two-stage method and the paper shows the convergence for mixed $\ell_1/\ell_2$. The improved empirical performance is also verified on the convex and nonconvex optimization problems. Especially, this work empirically verifies that the sparse solutions obtained by AdaHSPG+ are comparable to or better than those by Prox-SG, Prox-SVRG, and HSPG in the test performance for deep learning models on image datasets.

**Audience:**

Yes

**Broader Impact Concerns:**

Not applicable.

**Claims And Evidence:**

No

**Requested Changes:**

It would be nice if the authors could give the convergence rate analysis and could show the theoretical benefit of stochastic setting (i.e., sufficiently large $m_h, m_p$ with small batch sizes) against the almost deterministic setup ($m_h=m_p=1$). See weaknesses for the detail.

**Strengths And Weaknesses:**

**Strengths**

- This paper shows the convergence of AdaHSPG+ to a stationary point for the group-sparse regularization (i.e., mixed $\ell_1/\ell_2$ problem), whereas the convergence guarantee of HPSG has not been provided for this problem.

- AdaHSPG+ can find sparse solutions for deep learning models, maintaining the test performance achieved by the other methods such as Prox-SG, HSPG, and momentum method.

**Weaknesses**

- A major concern is a lack of convergence rate analysis shown for the other proximal methods. Because of this, we cannot see a certain separation from the deterministic method. Theorem 3.7 allows us to choose $m_h=m_p=1$. Then, the complexity per iteration under this setting is the same as the deterministic optimization methods (e.g., Prox-GD) because $|\mathcal{B}_k|=N$. Therefore, Theorem 3.7 cannot distinguish the performance between the stochastic update of AdaHSPG+ and the deterministic update setup above. And hence, I suspect that the convergence of AdaHSPG+ suggested by Theorem 3.7 basically relies on the deterministic update.
This point would be important to justify the claim that the proposed method does not suffer from randomness and does not struggle to obtain group-sparse solutions, which are raised as weaknesses of the past stochastic proximal methods in Section 1.2.
In addition, the lack of convergence rate analysis makes it impossible to suggest better hyperparameter settings to converge fast.
It would be nice if the authors could show the theoretical benefit of the proposed method based on the convergence rate analysis.

- Assumption 3.4 requires the boundedness of the iterations. Basically, this kind of a-posteriori assumption is not preferred. If the authors adopt this assumption, It would be better to give sufficient conditions or specific examples to satisfy it.

Minor comments:

- In Algorithm 2: Should $\mathcal{B}_k$ be added to the input of Algorithm 2 because it is used on line 2?

- In Algorithm 2: The indent of lines 10 and 11 should be corrected.

- In Experiment: How were groups of the fully connected layer in CNN models made? I could find an explanation for convolutional filters, but could not find it for fully connected layers.

- In the proof: The proof of the existence of $\sigma$ on page 15 is incorrect (maybe typos) and redundant.  If we make use of boundedness $||\nabla f_i(x)||\leq C$, then the variance can be immediately upper bounded by this.

---

> ### Comment · Reviewer_jnWn · 2023-04-14
> **Agree with concerns**
>
> Reviewer riu1 makes an important point. The proposed theory shows that there may (but may not) exist a nondeterministic version that converges-- yet the theory does not motivate taking this "risk" with a complexity benefit.

---

> ### Author Response · Authors · 2023-04-19
> **A revision incorporating the valued suggestions is available.**
>
> Dear Reviewer riu1,
>
> We appreciate your insightful questions and constructive suggestions.  We have incorporated all your valued suggestions into the revised manuscript. Please see our responses as follows. Look forward to further discussion.
>
> - **Lack of convergence rate analysis.**
>
> > Thanks for pointing it out. Please see our general response where we sincerely hope that could properly tackle your concern.
>
>
> - **When $m_h=m_p=1$, it is hard to distinguish the performance between the stochastic update of AdaHSPG+ and the deterministic variants of compared algorithms, for example Prox-SD.**
>
> > Thanks for the insightful comment. Upon close inspection of the proof for Lemma 3.6, the requirment on $m_p$ should be $b\geq m_p(m_p-1/2)$, where $b$ is the mini-batch size. The previous Algorithm 2 is stated in a way that the gradient over $\mathcal{B}_k$ computed during the  switching mechanism could be utilized at the beginning of each epoch. But the algorithm and theorem still works without usage of the gradient over $\mathcal{B}_k$. To avoid confusion, we have now revised the statement of Algorithm 2, which makes AdaHSPG+ is fully stochastic when $m_h=m_p=1$.
>
> - **Assumption 3.4 requires the boundedness of the iterations. Basically, this kind of a-posteriori assumption is not preferred. If the authors adopt this assumption, It would be better to give sufficient conditions or specific examples to satisfy it.**
>
> > This is a good question. In practice, Assumption 3.4 is typically satisfied in deep learning applications.  Sufficient conditions that imply this assumption may be more stringent than necessary. Based on numerous experiments with deep neural networks (DNNs), we have observed that the magnitude of each individual trainable variable is usually less than one when trained with standard optimizers like SGD and Adam. Moreover, using optimization techniques such as AdaHSPG+ or proximal methods can further decrease the magnitude of the variables due to the regularization term.
>
> - **How were groups of the fully connected layer in CNN models made?**
>
> > They are defined as the row of weighting matrix. We have added an explicit description in the revision.
>
> - **The proof of the existence of $\sigma$ on page 15 is incorrect (maybe typos) and redundant.**
>
> > Thanks for pointing them out. We have corrected the typos, which missed the $2$ in the exponent. We would leave the proof for clarity but are flexible to remove it upon the reviewer's request.
>
> Sincerely,
>
> Paper 837 Authors

---

> > ### Comment · Reviewer_riu1 · 2023-05-02
> > **Thanks for the response.**
> >
> > Thanks for the response and clarification.
> >
> > Additional question: [Chen et al. (2020b)] (HSPG paper) showed the convergence of HSPG without Lipschitz smoothness of the regularizer at the origin by excluding a small neighborhood of the origin. See Theorem 1 and the remark in [Chen et al. (2020b)]. Indeed, this exclusion deteriorates the Lipscthiz continuity and probably the theoretical convergence speed of HSPG, but it does not necessarily imply the separation between AdaHSPG+ and HSPG because of the lack of convergence rate. (I acknowledge that AdaHSPG+ does not require Lipschitz smoothness of the regularizer though.) In this sense, it would be nice to see a little more comparison and discussion with HSPG to justify the claim *"the analysis provided for HSPG does not extend to the mixed l1/l2 problem since itis not Lipschitz continuous in a neighborhood of the origin"*.
> >
> > Moreover, I'm not sure if AdaHSPG+ is truly a stochastic optimization method because Algorithm 3 (underlying solver) is simply non-stochastic proximal gradient descent when $b=1$ implying $m_p=1$.

---

> > > ### Author Response · Authors · 2023-05-03
> > > **Thanks for the follow-up questions.**
> > >
> > > Dear Review riu1,
> > >
> > > We appreciate your insightful follow-up questions. We have provided the responses below that we hope have addressed them adequately  and will incorporate into the final version. Look forward to further discussions.
> > >
> > > * **It would be nice to see a little more comparison and discussion with HSPG to justify the claim that "the analysis provided for HSPG does not extend to the mixed l1/l2 problem since it is not Lipschitz continuous in a neighborhood of the origin".**
> > >
> > > > Thanks for the insightful question. We would like to bring attention to the fact that the Lipschitz continuity assumption in the HSPG paper [Chen et al. (2020b)] is rather strong and overlooks a related case during the proof of sufficient decrease for the Half-Space step. This is the main reason why we assert that HSPG missed an asymptotic convergence result.
> > > > - For Theorem 1 to hold in HSPG, a rather strong assumption is required, which dictates that all iterates of the Half-Space step stay sufficiently close to the minimizer, i.e., $||x_{k}-x^*||<R$ almost surely (or at least with high probability). If this assumption is satisfied, $\nabla \Psi$ is Lipschitz continuous over $\mathcal{X}$, the region excluding an $\ell_2$-ball centered on the origin. However, this assumption is strong since the value of $R$ depends on the minimizer, which is typically unknown in practice beforehand, as well as the hyper-parameter $\epsilon$, which controls the aggressiveness of group sparsity exploration of the Half-Space step. In contrast, AdaHSPG+ does not impose such an assumption.
> > > > - Furthermore, the equation (22) in the [proof of Lemma 1 in [Chen et al. (2020b)]](https://arxiv.org/pdf/2009.12078.pdf) may not always be true if one Half-Space step projects at least one group of variables to zero. In such a case, the next iterate $x_{k+1}=x_{k} + \alpha_kd_k$ will have at least one group of variables as zero, which means that $x_{k+1}\not\in \mathcal{X}$. Therefore, $\nabla \Psi_{B_k}$ is not Lipschitz continuous at $x_{k+1}$, implying (22) invalid. As a result, the remaining proofs may require additional modifications to account for this missing case. Otherwise, Theorem 1 may not be true, as the assumption that all iterates of Half-Space steps satisfy $||x_{k}-x^*||<R$ would be violated.
> > >
> > > * **I'm not sure if AdaHSPG+ is truly a stochastic optimization method because Algorithm 3 (underlying solver) is simply non-stochastic proximal gradient descent when $b=1$ implies $m_p=1$.**
> > >
> > > > Thanks for the question. We would like to kindly clarify that even when $m_p=b=1$, Algorithm 3 is still considered stochastic. This is because line 5 in Algorithm 3 reduces to $v_{k,t} = \nabla f_i(\tilde x_{k,t}) - f_i(x_{k}) + \nabla f_{B_k}(x_k)$, where $i\sim \text{DiscreteUniform}[1,N]$ is used for sampling. Thus, even when $|B_k|=N$, i.e., the full batch, $v_{k,t}$ remains stochastic due to the sampling method. We also remark here that Algorithm 3 in our manuscript is modular and can be substituted by other algorithms that drive the first-order stationarity measure $||s(x_k;\eta)||$ to zero. For instance, one could use another variant of ProxSVRG proposed in [1].
> > >
> > > > [1] J Reddi, Sashank, et al. "Proximal stochastic methods for nonsmooth nonconvex finite-sum optimization." Advances in neural information processing systems 29 (2016).
> > >
> > > Sincerely,
> > >
> > > Paper 837 Authors

---

### Review · Reviewer_jnWn · 2023-04-10

**Summary Of Contributions:**

The authors address the challenge of combining group-sparse regularization with stochastic optimization algorithms, which is famously challenging (challenging to have convergence guarantees, be computationally efficient, and provide group-sparse solutions). The authors propose a new algorithm that has faster convergence than a previously published algorithm due to integration of a variance reduction technique with a novel strategy for adaptively estimating the support of a potential solution.

The algorithm is iterative and at each iteration, one of two types of steps is performed based on an automatic selection method. Specifically, the set containing groups that are both predicted to remain nonzero after a batch prox-gradient step and are sufficiently far from the origin, is isolated. The prox-gradient steps for that and its complementary partition are computed; whichever partition's corresponding step is larger in magnitude is the set of variables that is focused on (how much "larger" being tunable by an algorithm parameter). If the former (support group) can yield a larger step then the Enhanced Half-Space (EHS) step is activated to explore whether those groups should remain nonzero. Otherwise some steps of Prox-SVRG are performed (proximal stochastic variance-reduced gradient method).


Proofs are provided to show that under specific conditions,
* EHS decreases the objective function
* Prox-SVRG decreases the objective function if batch-size is sufficiently large
* A stationary point will be achieved by the overall algorithm if one exists. Asymptotic convergence is proved, while complexity bounds are not established (although insights are to provided for how they might be, it is avoided due to practical performance limitations).

The algorithm is demonstrated to outperform its primary competitor (HSPG: half space projected gradient algorithm) and their consitutent algo predecessors (proxSG, proxSVRG) on convex and non-convex problems. Performance is compared in terms of loss function value, group-sparsity ratio, and accuracy. Visual comparison of metrics as they evolve with training is provided for non-convex (deep neural network training) examples.



**Audience:**

Yes

**Broader Impact Concerns:**

I don't have any concerns regarding ethical implications.

**Claims And Evidence:**

Yes

**Requested Changes:**

# Critical to securing my recommendation:

1. At times it seems clear that $N$ is the number of samples, but the summation over $f_i(x)$ also runs to $N$ in multiple places. I think you need a different variable the number of functions and number of samples... I think it would go a long way to denote an explicitly stochastic version of (1) in the same section.

2. SAG is not defined in lit review (Sec 1.2)

3. Above Eqn (2), you claim that Prox_r has a closed-form solution. However, you don't provide this formula and the reference you cite there does not either. I think since $\mathbf{x}$ is in disjoint groups it's just soft thresholding applied to each $||\mathbf{x}_g||$? I suggest you just say what it is (or provide a more specific reference).

4. A hyperparameter sweep for more comprehensive comparison (see 2nd point in Weaknesses)

# Simpler suggestions:

## Sec 1.1
* Perhaps one more sentence explaining pros/cons of overlapping vs non-overlapping group sparsity. i.e., time-invariance, sensitivity to group-size
* It might be nice to have a sentence explaining why GS is referred to as L1/L2 norm (i.e., because it's equivalent to the L1 norm of a vector of L2 norms, and hence induces sparsity on that vector, removing some groups and keeping others).
## Sec 1.3
* In the first bullet point on improvements, please explain more explicitly what is the difference between "an expectation-type result" and "a pure expectation-type result"
## Sec 2.1
* paragraph break before "The *switching mechanism...*"
* Also, that sentence could be rewritten backward to increase readibility.. i.e., explain first... "The switching mechanism proceeds according to which set of variables, I^HS or I^NHS, is expected to yield more progress after a step of optimization, via EHS or PSVRG, respectively. This is gauged by their size..." (or so). All that being said, is it trivial as to why the energy of the partition indicates potential improvement (more energy $\Rightarrow$ more possible improvement) or is there more to it that is going unexplained? One sentence of explanation would go much further than "it makes sense".
* Something is wrong with your quotes around "projection" above Eqn (5)
* Accidental indentation in line 12 of Algorithm 2?
## Sec 4.2
* Since there is little effective difference between HSPG and AdaHSPG+ in the convex setting, it might be more useful to show results for a setting or metric that reveals a difference (e.g. 50 epochs, or "epochs to convergence").
## Sec 4.3
* Why isn't there a comparison with Adam? This optimizer is a go-to for many/most ML publications.

**Strengths And Weaknesses:**

# Strengths
* Excellent/clear explanation (with illustrations) of the algorithmic mechanisms
* Introduced hyperparameters are intuitive and their affects are discussed (except the sensitivity of kappa)
* Overcoming the limitation of HSPG that once it zeroes out a group, it does not get re-updated.
* Little to no effect on wallclock runtime
* Good breadth of experiments

# Weaknesses
* A notation error regarding $N$ (unless I am just confused)
* For the convex problem: hyperparameter sweeps for the convex methods seem necessary for an interesting comparison. Table 1 seems to only verify that they converge. When do the two main competitors differ? Maybe a robustness test would be interesting.
* Although justification is given, complexity analysis is not hashed out. It's probably better to do this anyway, even if in the numerical experiments you explore the practical tweak (variance reduction in EHS)

---

> ### Author Response · Authors · 2023-04-19
> **A revision incorporating the valued suggestions has been uploaded.**
>
> Dear Reviewer jnWn,
>
> We appreciate your insightful comments and constructive suggestions. The suggestions have been made in the revised manuscript. Please see our responses to the comments. Look forward to further discussion.
>
> - **At times it seems clear that $N$ is the number of samples, but the summation over $f_i(x)$ also runs to $N$ in multiple places. I think you need a different variable the number of functions and number of samples. I think it would go a long way to denote an explicitly stochastic version of (1) in the same section.**
>
> > This is a good question. Our objective function (Formulation (1)) can be viewed as a generalization for the objective of emperical risk minimization problem, where $N$ denotes the number of samples. Consider a set of $N$ data points $\{\xi_i\}_{i=1}^{N}\subseteq\mathcal{D}$ and a loss function $\ell: \mathbb{R}^n \times \mathcal{D} \to \mathbb{R}$, by setting $f_i(x):=\ell(x,\xi_i)$ the loss with respect to the $i$th data point and any model parameter $x \in \mathbb{R}^n$.
>
> > Our target objective function $\frac{1}{N}\sum_{i=1}^{n}f_i(x)=\frac{1}{N}\sum_{i = 1}^{n}\ell(x,\xi_i)$ then becomes the standard finite-sum objective of emperical risk minimization problem. We hope this clarification resolves the confusion.
>
> - **SAG is not defined in lit review (Sec 1.2)**
>
> > Thanks for pointing it out. We have added the full description regarding SAG in Sec 1.2.
>
> - **Above Eqn (2), you claim that Prox_r has a closed-form solution. However, you don't provide this formula and the reference you cite there does not either. I think since $x$ is in disjoint groups it's just soft thresholding applied to each $\|\bf{x}_g\|$? I suggest you just say what it is (or provide a more specific reference).**
>
> > Yes, it admits a closed form when groups are disjoint, i.e, $\text{prox}_{\eta \|\cdot\|}({\bf{x}}_g)=(1-\frac{\eta}{\max\{\|\bf{x}_g\|, \eta\}})\bf{x}_g$. We have made the suggested modifications.
>
> - **Comment on Sec 1.3: please explain more explicitly what is the difference between "an expectation-type result" and "a pure expectation-type result"**
>
> > **By an expectation-type result**, we refer to the $1 - \tau$ lower probability bound in Theorem 1 of the **HSPG** paper [Appendix, 1]. Intuitively, since HSPG permanetely fixes groups it identifies should be zero, its convergence requires all zero groups it identifies to be correct, which can be shown to hold with at least probability $1 - \tau$.
>
> > This probabilistic result is because of the two-stage design of HSPG that permanently fixes zero groups at zeros, thereby its convergence result necessitates the idendification of zero groups being all correct. In contrast, in AdaHSPG+ we address this weakness by  designing an adaptive switching mechanism to correct false positive zero groups.
>
> > **By a pure expectation-type result**, we refer to Theorem 3.7 in this paper that states $\liminf_{k\to\infty} \mathbf{E}[\|\bf{s}(\bf{x}_k;\eta)\|] = 0$. This is a standard (asymptotic) convergence in expectation result.
>
> - **Comment on Sec 4.3: Why isn't there a comparison with Adam? This optimizer is a go-to for many/most ML publications.**
>
> > This is a good question. We did not compare with Adam is because the benchmark image classification problems included in the papers are typically trained via SGD in other literatures. Therefore, we included the comparison with the full models trained with SGD in Table 2, where our AdaHSPG+ optimizer can reach competitive validation accuracy **and** high group sparsity, which gives benefits in practice by delivering slimmer model to be deployed onto edge devices. We are currently conducting an additional super-resolution experiment, which baseline full model is trained via Adam and will include it into the final version.
>
>
> Sincerely,
>
> Paper 837 Authors

---

> > ### Comment · Reviewer_jnWn · 2023-04-24
> > **critical suggestions addressed**
> >
> > My critical suggestions are addressed. Thanks to the authors

---

### Review · Reviewer_dnVZ · 2023-04-22

**Summary Of Contributions:**

This manuscript proposes a new stochastic optimization methods for structured $\ell_1$ minimization problems. The authors based their method by combining previous ideas while leveraging a new switching strategy, providing asymptotic convergence in expectation for convex problems. Empirically, the authors demonstrate their algorithm in convex and non-convex problems, showing superior performance in terms of the obtained degree of sparsity - particularly in the non-convex application of training sparse neural networks.

**Audience:**

Yes

**Broader Impact Concerns:**

No concerns

**Claims And Evidence:**

Yes

**Requested Changes:**

## Main concerns:
* I have some trouble understanding some parts of this algorithm. More specifically, while the authors do explain that some of the benefits of their approach relies on replacing the 2-parts algorithm by an automatic switching strategy, is not quite clear why this is indeed beneficial. Moreover, it is also unclear to me why the projection step in Eq 5 is needed; naturally, probably this is clear to those readers who are already very familiar with HSPG, but I can't pin exactly to the conceptual reason this step is beneficial to the minimization of Eq. (1).x

* I find some statements throughout the manuscript exaggerated. For example, the authors refer to their "significant improvements" over HSPG, which might be true for the improved sparsity, but the presented converge result is weaker (it's only an asymptotic result). In other places (e.g. page 2) the authors mention that HSPG depends on heavily tuning some hyper-parameters, alluding to the fact that their method seemingly gets around this problem. Yet, the proposed method also has a large number of hyperparameters that need to be manually defined, and seem to arbitrary values (see Implementation details in Sect 4.1).

* The results reported in the tables clearly indicate that the presented method is superior in terms of the final achieved sparsity (at least in the non-convex case). However, once looking at the training curves in Fig 2 and 3, these are often very erratic, and the final numbers clearly depend on how long the optimization algorithm is run for. This weakens the observation that their results are homogeneously better.

* Notation and nomenclature needs to be improved. Very often $x$ and $\mathbf{x}$ are used interchangeably (see e.g. Eq 3). Moreover, what the authors refer to as "proximate gradient step" (in Eq 2) is in fact a gradient mapping operator - please see [Beck, Amir. First-order methods in optimization, 2017,] Chapter 10. The gradients $\nabla_g\psi_\mathcal{B}(x)$ are confusing as it seems that the gradient is taking w.r.t. the assignment of the group, $g$.

* There authors often generally mention that, unlike other methods, theirs obtains a better degree of sparsity (E.g. in Sec 1.2). I would encourage the authors to stress that this is true in the non-convex case. In the case convex, the solution is typically unique whp, and so the degree of sparsity depends not on the algorithm but on value of the regularization parameter.

* I'm unsure about Assumption 3.4. More precisely, I understand why this is needed, but it seems to me that the authors should be able to derive this condition under assumptions 1-3, employing the fact that the projection is a non-expansive map and that the step size is chosen appropriately. Thus, is it really necessary?

* In commenting on Lemma 3.5, the authors state that if the stepsizes are chosen small enough, and the batch size sufficiently large, then EHS decreases the function value. Yet, from the expression in Lemma 3.5, I believe it's also true that, regardless the value of the batch size, this can be made to decrease if $\alpha_k$ is chosen small enough, no?

* Commenting on their experiments, the authors mention in passing that the provided relative runtimes (in appendix) shows that their methods is competitive w.r.t HSPG. It is true that the runtime is in the same order of magnitude, but it's also true that the runtime is consistently higher. The authors should state this, and moreover, comment on the sources for this (slight but consistent) increase.

* The authors could make their conclusions more precise, by instead of saying "... AdaHSPG+ significantly outperformes HSPG [..] in terms of various performance metrics...", saying that the improvement is obtained in terms of the resulting sparsity in non-convex problems, at the expense of slightly different theoretical guarantees and slightly increased runtime.

## Smaller concerns
* Use of  "quotations'' should be fixed, as they are consistently incorrect - use `` and ' '.
* Abstract: "employed each iteration" -> employed at each iteration.
* The first paragraph in the introduction is all about the use of structured sparsity in machine learning. Yet, referenced literature is only from 2016 onward. This disregards a huge amount of work on Lasso, Group Lasso, Hierarchical Lasso and its variants. I understand that the objective of this work is different from those contributions, but I find it incorrect to refer to these topics and not cite these contributions.
* SAG seems undefined on Page 2?
* "HSPG [...] clearly has limitations". It's unclear to the reader why it is specifically stated that "clearly" HSPG has limitations, particularly because all methods have limitation. Just state what those limitation are.
* I think $\nabla_\mathcal{I}^{grad}$ in Lemma 3.5 seems undefined, as it's defined later. It'd be much clearer if its definition was moved up.
* Missing capitalization before Theorem 3.7
* what is $\lim \inf_{k\to\infty}$ in Them. 3.7?
* "stand-a-alone" -> "stand-alone"


**Strengths And Weaknesses:**

## Strengths

* To my knowledge, the methodology is novel, and the problem that the authors study is interesting.
* The numerical comparison looks appropriate, and the improvement in terms of obtained sparsity (in the non-convex case) seems significant.

## Weaknesses

* I'm afraid there are details that I still don't fully grasp, in part (probably) because of a lack of clarity, notation inconsistencies lack of justification of specific steps (see below).
* Given the results presented in the paper, I believe the qualitative description of these are often exaggerated (see below).

---

> ### Author Response · Authors · 2023-04-22
> **A revision incorporating the constructive suggestions has been uploaded.**
>
> Dear Reviewer dnVZ,
>
> We appreciate your insightful comments and constructive suggestions. Your suggested changes to improve our presentation have been made in the revised manuscript. Please see our responses to your questions that hopefully address them adequately. Look forward to further discussion.
>
> - **The benefits of automatic switching strategy.**
>
> > The automatic switching strategy in AdaHSPG+ overcomes two design limitations of HSPG. First, HSPG requires the users know when switching from the first Prox-SG stage to the second Half-Space stage which typically requires task specific hyper-parameter fine-tuning. Second, since Half-Space stage in HSPG would permanently fix zero groups as zero, it may yield false positive group sparsity and lack capacity to correct in accordance. The automatic switching mechanism in AdaHSPG+ addresses the two limitations and benefits both theorem and practice.
>
> - **Why Half-Space projection is needed?**
>
> > Please refer to our general response to all reviewers and AEs where we present the motivations of Half-Space projector which is designed to address the limitations of proximal methods over group sparsity exploration in deep learning problems. In the final version, we will add more descriptions regarding HSPG.
>
> - **The presented converge result is weaker than HSPG.**
>
> > We kindly argue that AdaHSPG+ equips with a stronger convergence result in asymptotics than HSPG (which is missing). In fact, AdaHSPG+ overcomes two theoretical limitations in HSPG: (i) HSPG requires Lipschtiz continuity of the gradient of the sparsity-promoting regularizer, which does not hold for the regularizer considered in this work, and (ii) HSPG requires the zero-groups identified to be correct (this is a significant assumption) since HSPG proceeds to **permanently** fix those groups at zero.
>
> - **The authors mention that HSPG depends on heavily tuning some hyper-parameters, alluding to the fact that their method seemingly gets around this problem. Yet, the proposed method also has a large number of hyperparameters that need to be manually defined, and seem to arbitrary values.**
>
> > Thanks for the question. In general, HSPG typically requires fine-tuning (i) the $\epsilon$ in Eq. (5) and (ii) when to switch from the first phase into the second phase. $\epsilon\in [0,1)$ controls the aggressiveness of group sparsity exploration of the Half-Space projector. Higher $\epsilon$ typically yields higher group sparsity. The switch between two phases affects the convergence of HSPG a lot. Both hyperparameters typically require task specific fine-tuning efforts. AdaHSPG+ designs adaptive mechanisms for $\epsilon$ and the switch to largely avoid such efforts. The adaptative switching mechanism further results in the asymptotic convergence of AdaHSPG+ which is missing from HSPG.
>
> > In addition, we admit that AdaHSPG introduces new hyper-parameters related to the adaptive mechanisms but would like to highlight that we largely follow the common conventions to set them up. For example, the $\mu$ in the adaptive switching mechanism is set as 1 to equally favor the occurrence of both ProxSVRG step and enhanced half-space step. Adaptively increasing or decreasing $\epsilon$ by a factor as $2$ follows the standard linear-search type strategy in optimization. During our experiments, the adaptive mechanisms delivered more benefits than the efforts of setting them up.
>
> Sincerely,
>
> Paper 837 Authors

---

> > ### Author Response · Authors · 2023-04-22
> > **Continued responses**
> >
> > - **Once looking at the training curves in Fig 2 and 3, these are often very erratic, and the final numbers clearly depend on how long the optimization algorithm is run for. This weakens the observation that their results are homogeneously better.**
> >
> > > This is a great question. We would like to kindly highlight that the curves of group sparsity are **not** erratic but meet our **goal** of AdaHSPG+ instead. In particular, we design AdaHSPG+ to adaptively switch between Prox-SVRG step and enhance half-space step and adaptively update the $\epsilon$ upon the optimality of the current iterate. These adaptive mechanisms would affect the group sparsity level in accordance. In contrast, HSPG is a two-stage method which will permanently fix zero groups as zero, then may predict false positive group sparsity and can not deliver satisfactory theoretical convergence results. Meanwhile, the results in Table 1-3 are reported by taking average of several independent repeats. Therefore, we would consider the curves of group sparsity as valuable evidence to demonstrate the effectiveness of the algorithmic design of AdaHSPG+.
> >
> > > In addition, the group sparsity curves of AdaHSPG+ consistently exhibit frontier than proximal methods on 5 out of 6 image classification experiments during the whole training process except VGG16 on CIFAR10. For VGG16 on CIFAR10, AdaHSPG+ outperforms proximal methods around the 225th epoch when the learning rate decreased to $10^{-4}$ from $10^{-3}$ till the final convergence. The rise of group sparsity around the 225th epoch of AdaHSPG+ is caused by the iterate being more closer to the optimum under a smaller learning rate. Then the $\epsilon$ is adaptatively updated to yield more group sparsity, which also meets the Theorem 3.8 in the revision regarding group sparsity identification. Meanwhile, we follow the benchmark learning rate schedulers as other literature for training the neural networks. In the end, we thank the reviewer for the great question again but would like to kindly highlight that such group sparsity evolutions are what we are **seeking** for the algorithm.
> >
> > - **What the authors refer to as "proximal gradient step" (in Eq 2) is in fact a gradient mapping operator - please see [Beck, Amir. First-order methods in optimization, 2017,] Chapter 10. The gradients $\nabla_g\psi_\mathcal{B}(x)$ are confusing as it seems that the gradient is taking w.r.t. the assignment of the group, $g$.**
> >
> > > We kindly point out Eq (2) is different from the proximal operator as it is defined as the **difference** between the output of the proximal operator and the current iterate $x$. And this is the reason we denote $s$ as the proximal gradient step. $\nabla_g\psi_\mathcal{B}(x)$ indeed means the partial gradient with respect to the group $g$. And this quantity is used Eq (3) to decide what groups of variables are updated by the enhanced half-space step.
> >
> > - **There authors often generally mention that, unlike other methods, theirs obtains a better degree of sparsity (E.g. in Sec 1.2). I would encourage the authors to stress that this is true in the non-convex case. In the case convex, the solution is typically unique whp, and so the degree of sparsity depends not on the algorithm but on value of the regularization parameter.**
> >
> > > Thanks for the suggestion. In the revision, we have moved the convex experiments into the appendix and focused on non-convex experiments in the main body.
> >
> > - **Assumption 3.4 seems to be redundant as it is implied by the assumptions 1-3, the projection is a non-expansive map, and that the step size is chosen appropriately.**
> >
> > > This is a great question. We need Assumption 3.4 since the Half-Space projection is not a standard Euclidean projection, so that we could not show the non-expansiveness.
> >
> > - **From the expression in Lemma 3.5, I believe it's also true that, regardless the value of the batch size, this can be made to decrease if $\alpha_k$ is chosen small enough, no?**
> >
> > > Yes, you are right.
> >
> > - **The authors could make their conclusions more precise, by saying that the improvement is obtained in terms of the resulting sparsity in non-convex problems, at the expense of slightly different theoretical guarantees and slightly increased runtime.**
> >
> > > Please refer to our response regarding (i) convergence result that AdaHSPG+ equips with a **stronger** convergence result than HSPG (which is missing); and (ii) fewer hyper-parameter tuning efforts of AdaHSPG+ due to the adaptive mechanisms.
> > Meanwhile, we have rephrased the conclusion section to make it more clear.
> >
> > - **What is $\liminf_{k\to\infty}$ in Thm 3.7.**
> >
> > > `liminf` stands for [lim inferior](https://en.wikipedia.org/wiki/Limit_inferior_and_limit_superior). This theorem can be interpreted as that in expectation there is a subsequence of the $\|s(x_k;\eta)\|$ converges to 0, where $\|s(x_k;\eta)\|$ is the measure of the first order stationarity.
> >
> > Sincerely,
> >
> > Paper 837 Authors

---

> > > ### Comment · Reviewer_dnVZ · 2023-04-23
> > > **Almost all comments addressed**
> > >
> > > Dear authors,
> > >
> > > Thank you for promptly responding to my questions, which have for the most part clarified my doubts. A couple of follow-ups:
> > >
> > > * **Definition of gradient step**: I think the authors might have misunderstood my previous comment. What the authors defined as "proximal gradient step" is indeed known as (the negative) ***gradient map***: $G(x) = x - \text{Prox}(x)$; please see Definition 10.5 in [Beck, Amir. First-order methods in optimization, 2017].
> > >
> > > * **Notation for gradients** What I meant in the previous comment is that I don't think that the notation $\nabla_g\psi(x)$ computes the gradient w.r.t. the assignment of the group denoted by $g$, but instead computes the gradient w.r.t. $x_g$. Is this correct? If so, $\nabla_g\psi(x)$ is short-hand notation for $\nabla_{x_g}\psi(x)$?
> > >
> > > * **Comment on why iterative prox-grad fails in non-convex problems** I welcome the explanation that the authors provided in the general response on why iterative prox-grad method fail to achieve considerable sparsity. I'm slightly confused about this, however, as it seems like this is simply the result of the lack of appropriate scaling of the strength of the proximal operator? Maybe I'm missing something.
> > >
> > > * **Minor Comments** I welcome the authors' rephrasing of some of the statements in the conclusion, as well as the new experiments. Please do note some other minor/trivial observations that remain to be fixed, such as use of quotations.

---

> > > > ### Author Response · Authors · 2023-04-24
> > > > **A new revision incorporating follow-up comments has been uploaded.**
> > > >
> > > > Dear Reviewer dnVZ,
> > > >
> > > > We appreciate your valuable follow-up suggestions and questions. We are pleased to see that our first-round responses have addressed most of your concerns. We are now providing responses to the follow-up questions below. Meanwhile, a revised manuscript incorporating the valuable suggestions has been uploaded. Look forward to further discussion.
> > > >
> > > > - **Definition of gradient step**
> > > >
> > > > > Thanks for the clarification. You are right. We have added description to point out the connection in page 3.
> > > >
> > > > - **Notation for gradients**
> > > >
> > > > > Yes, you are right. For clarification, we have added one more equation to explicitly present it, i.e., $\nabla_g \psi(x)=\frac{\partial \nabla \psi(x)}{\partial [x]_g}$ on page 3.
> > > >
> > > > - **Comment on why iterative prox-grad fails in non-convex problems.**
> > > >
> > > > > This is a great question. You are right that proximal methods could yield reasonable group sparsity by increasing the regularizer coefficient $\lambda$ in the target problem $minimize_{x\in\mathbb{R}^n} f(x) +\lambda r(x)$. However, a larger $\lambda$ tends to shift the goal of minimizing $f$ to minimizing the regularizer, which deteriorates model performance instead. Consequently, proximal methods are typically hard to achieve **both** high group sparsity and high model performance simultaneously. The underlying reasons are due to their group sparsity exploration heavily relying on sufficiently large regularizer coefficient $\lambda$ and learning rate $\alpha$, while the performance of neural networks are easily degraded under large $\lambda$ and $\alpha$.
> > > >
> > > > > HSPG-type methods address such limitations via a novel Half-space projection which group sparsity exploration much less sensitive on these two hyper-parameters. As a result, under similar group sparsity levels, we could observe that proximal methods typically have worse model performance than HSPG-type methods, e.g., the Bert on SQuAD experiment.
> > > >
> > > > - **Minor Comments**
> > > >
> > > > > Thanks for the suggestions. We carefully revisited the minor comments and incorprated them into the revision. In particular, we have
> > > > >- Fixed typos, quotations, and statements.
> > > > >- Added more references into the first and second paragraphs in introduction.
> > > > >- Moved up the definition of $\mathcal{I}_{k,t}^{Grad}$ onto page 5.
> > > >
> > > > Sincerely,
> > > >
> > > > Paper 837 Authors

---

> > > > > ### Comment · Reviewer_dnVZ · 2023-04-24
> > > > > **Thank you for your responses**
> > > > >
> > > > > I thank the authors for addressing all of my comments.

---

### Author Response · Authors · 2023-04-14
**Update from the authors**

Dear reviewers and AEs,

We greatly appreciate your constructive comments and valued suggestions. For a quick update, we are conducting more non-convex experiments, preparing our responses to the raised questions and concerns and expecting to post them in the early of next week.

Thanks for your consideration. Have a nice weekend.

Sincerely,
Paper 837 authors

---

### Author Response · Authors · 2023-04-19
**Thank all the reviewers and AEs for the insightful comments. Look forward to further discussion.**

Dear reviewers and AEs,

We deeply appreciate all the insightful comments and constructive suggestions that have helped us improve our manuscript. In our revision, we have carefully addressed each comment and highlighted our changes in blue text. Below, we present our algorithm's motivations and our response to the general questions regarding convergence rate and hyper-parameter tuning. We hope that our responses adequately address the reviewers' concerns and look forward to further discussion.

- **Why we studied the HSPG method rather than a proximal method?**

> **Defectiveness of proximal method in deep learning (DL) tasks.** Proximal methods have been used successfully to solve sparse optimization problems in classical machine learning problems. However, in the current era of deep learning, models and tasks have become much more complex, posing significant challenges to proximal methods. Although proximal methods still perform well in terms of objective function convergence, they often fail to produce sparse solutions. Their failure is due to the reliance on the proximal operator with projection radius $\alpha\lambda$ to yield (group) sparsity, where $\alpha$ is the learning rate, and $\lambda$ is the regularization coefficient in the following target problem: $\text{minimize}_{x} \ f(x)+\lambda r(x)$.  The projection radius $\alpha\lambda$ vanishes to zero since $\lambda$ is typically $\ll 1$, and the learning rates for neural networks are typically vanishing as well. As a result, proximal methods do not have sufficient capacity to practically generate sparse solutions in DL problems.

> **HSPG-type methods are proposed to better explore group sparsity.** To address these limitations, it became necessary to design a new sparse optimizer that could explore sparsity without relying too heavily on the learning rate and $\lambda$. HSPG was introduced with this motivation, featuring a Half-Space projector that provided a novel mechanism for producing group sparsity. In practice, HSPG-type methods have shown noticeably better results compared to proximal methods, particularly in deep learning tasks, i.e., maintaining competitive objective function convergence while achieving significantly higher group sparsity. We developed AdaHPSG+ (the proposed algorithm in our paper) to further refine the algorithm and provide theoretical guarantees (missing from the HSPG work).

- **Lack of convergence rate.**

> We thank the reviewers for this comment, and we  recognize the benefit of presenting a convergence rate for optimization algorithms.  With that said, we would like to emphasize that the asymptotic convergence of AdaHSPG+, the best result we could obtain so far, is already a significant improvement compared to HSPG. AdaHSPG+ overcomes two theoretical limitation in HSPG: (i) HSPG requires Lipschtiz continuity of the gradient of the sparsity-promoting regularizer and (ii) HSPG requires the zero-groups identified in its first phase to be correct (this is a signficant assumption) since HSPG proceeds to **permanently** fixe those groups at zero for the remainder of its half space computations.

> We also admit that, at this time, we do not know how to prove a convergence **rate** for AdaHSPG+ due to its more sophisticated structure compared to other commonly studied algorithms, such as stochastic proximal gradient variants.  In our work, we focus on achieving a better sparsity capacity (while also maintaining a competitive pratical objective convergence and proving a convergence result).  To validate our approach, we conducted deep learning experiments covering both convolutional neural networks and large-scale transformers with comprehensive ablation studies in the revision. We further provide one new theorem (Theorem 3.8) that highlights the potential for superior group sparsity identification of AdaHSPG+ compared to proximal methods. We sincerely hope that our responses have adequately addressed the reviewers' concerns.

- **Hyper-parameter fine-tuning.**

> We thank all reviewers for the valuable suggestion. In the revision, we sweep two main hyper-parameters of proximal methods, i.e., learning rate $\alpha$ and regularizer coefficient $\lambda$. We first fine-tune the $\lambda$ in the range {$10^{-2}, 10^{-3}, 10^{-3}$}. It can be seen from Figure 2 in the revised manuscript that $\lambda=10^{-3}$ strikes a good balance between accuracy on group sparsity. AdaHSPG+ performs better than ProxSVRG and ProxSG by consistently exhibiting the frontier of Patero curves of group sparsity and validation accuracy. We then fix $\lambda=10^{-3}$, and fine-tune stepsize $\alpha$ with different updating strategies and initial values. It can be seen from Figure 3 in the revised manualscript that AdaHSPG+ outperforms proximal methods again over all learning rate variations. Both results validate that HSPG methods outperform proximal methods for better group sparsity and competitive objective.

Sincerely,

Paper 837 Authors

---

> ### Author Response · Authors · 2023-04-19
> **Highlight of Revision**
>
> **Highlight of Revision**
>
> We incorporated all our responses into the revision. The main changes are highlighted as follows.
>
> - **One more non-convex experiment over large-scale transformer.**
>
>    > We compared with ProxSSI and significantly outperform it in terms of both group sparsity and model performance.
>
> - **Comprehensive hyper-parameter tuning**.
>
>    > We conducted two ablation studies over the learning rate and the regularizer coefficient separately. Results show AdaHSPG+ consistently perform better than proximal methods regarding sparsity level and maintain competitive
>
> - **One more theorem regarding group sparsity identification.**
>
>    > Theorem 3.8 to present the superiority of AdaHSPG+ than Prox-SG in terms of group sparsity identification.
>
> We thank all reviewers and AEs again for the considerations of our work!
>
> Sincerely,
>
> Paper 837 Authors

---

### Decision · Action_Editors · 2023-05-12

**Recommendation:** Accept as is

**Comment:**

The authors combine a group-sparse regularization with stochastic optimization algorithms, and employ a new switching strategy. They also provided an asymptotic convergence in expectation for convex problems. After the revision, the authors also provide sufficient numerical experiments to support using their algorithm, and all the reviewers agreed the paper should be accepted.

**Audience:**

The audience is geared towards the ML community interested in Optimization and sparsity.

**Claims And Evidence:**

After the revision, the claims now match the evidence given.